# Two decades of pH$_T$ measurements along the GO-SHIP A25 section in the North Atlantic

Fiz F. Pérez[1], Marta López-Mozos[1,2*], Marcos Fontela[1], Maribel I. García-Ibáñez[3], Noelia Fajar[4], Xosé Antonio Padín[1], Mónica Castaño-Carrera[1], Mercedes de la Paz[1], Lidia I. Carracedo[5], Marta Álvarez[6], Herlé Mercier[5], Pascale Lherminier[5], Antón Velo[1*]

[1]Instituto de Investigaciones Marinas (IIM), CSIC, Vigo, 36208, Spain
[2]Facultad de Ciencias del Mar, Universidade de Vigo, Spain
[3]Centro Oceanográfico de Illes Balears (COB-IEO), CSIC, Palma, Spain
[4]Centro Oceanográfico de Vigo (COV-IEO), CSIC, Vigo, Spain
[5]University of Brest, CNRS, Ifremer, IRD, Laboratoire d'Océanographie Physique et Spatiale (LOPS), IUEM, Plouzané, 29280, France
[6]Centro Oceanográfico de A Coruña (COAC-IEO), CSIC, A Coruña, Spain

*Correspondence to*: Marta López-Mozos (mlopezm@iim.csic.es) and Antón Velo (avelo@iim.csic.es)

## Abstract

The North Atlantic (NA) GO-SHIP (Global Ship-based Hydrographic Investigation Program) A25 OVIDE-BOCATS (Observatoire de la variabilité interannuele à décennale en Atlantique Nord - Biennial Observation of Carbon, Acidification, Transport and Sedimentation in the North Atlantic) section is a long-term repeat hydrographic transect extending from Portugal to Greenland. Since 2002, physical and biogeochemical measurements have been carried out biennially along the OVIDE-BOCATS section, contributing to a better understanding of water mass properties, mixing, circulation, carbon storage, and climate change impacts such as ocean acidification (OA) in the NA. In particular, the high-precision pH measurements on the total hydrogen ion scale (pH$_T$) from the OVIDE-BOCATS program represent a key milestone in monitoring OA in this particularly climate sensitive region. The method used for pH$_T$ determination relies on adding meta-cresol purple (mCP) dye to the seawater sample and spectrophotometrically measuring its absorbances at specific wavelengths. The OVIDE-BOCATS program has used unpurified mCP dye, which impurities have been proven to bias pH$_T$ values. Here we quantified the bias induced by these impurities in pH$_T$ measurements. We found that measurements carried out using the unpurified mCP dye tend to be, on average, **0.011 ± 0.002** pH$_T$ units higher than those obtained using the purified mCP dye, with this difference slightly decreasing at higher pH$_T$ values. Moreover, we tested independent methods to correct the effect of impurities in both the historical and recent OVIDE-BOCATS pH$_T$ data, demonstrating that the correction is consistent across methods. The long-term pH$_T$ dataset has been updated to include newly acquired data and absorbance measurements, and to standardize corrections for mCP dye impurities. This effort results in a twenty-year dataset of pH$_T$ corrected for mCP dye impurities, that demonstrates the possibility of a global effort to improve the reliability and coherency of spectrophotometric pH$_T$ measurements made with unpurified mCP dye. The corrections applied to our pH$_T$ dataset have negligible implications for the OA rates previously reported, but they do affect the depth of the aragonite saturation horizon, implying a shoaling of approximately 150 m. The dataset is available at https://doi.org/10.5281/zenodo.17789895 (Pérez et al., 2025).

# 1 Introduction

The oceanic absorption of anthropogenic $CO_2$ ($C_{ant}$) is causing major changes in the marine carbonate system chemistry (Friedlingstein et al., 2023; Le Quéré et al., 2015). The ocean is slightly basic generally; however, $C_{ant}$ uptake increases the concentration of total hydrogen ions ($[H^+]_T$), decreasing pH and the concentration of carbonate ions. These changes are collectively referred to as ocean acidification (OA) (Caldeira and Wickett, 2003; Orr et al., 2005), and they are especially detrimental for calcifying marine organisms and their ecosystems (IPCC, 2019). OA is a major concern for decision-makers at both local and global scales due to its potential impacts on marine ecosystem health and food security (Gattuso et al., 2015). The future impact of OA will depend on variations in the long-term mean and the short-term temporal variability of the marine carbonate system (Kwiatkowski and Orr, 2018).

Due to their physicochemical characteristics, surface waters in polar and subpolar regions are expected to experience the greatest OA impact (IPCC, 2019; Orr et al., 2005). However, the impact of OA is not limited to surface waters. Recent observations have shown that intermediate layers of the North Atlantic (NA) are experiencing higher OA rates than surface waters (Pérez et al., 2021; Resplandy et al., 2013) due to its distinctive circulation dynamics. The upper limb of the Atlantic Meridional Overturning Circulation (AMOC) transports $C_{ant}$ from the subtropics to the Subpolar North Atlantic (SPNA), where it is transferred to intermediate and deep layers—with lower buffering capacity than surface layers—through deep winter convection and water mass formation (Asselot et al., 2024). This process would ultimately contribute to the deterioration of NA deep-water coral ecosystems (Fontela et al., 2020a; García-Ibáñez et al., 2021; Gehlen et al., 2014; Perez et al., 2018).

The demand for open-access OA data is increasing, driven by United Nations Sustainable Development Goal (SDG) 14 and by its role as a climate indicator recognized by the World Meteorological Organization (WMO). Although great progress in autonomous marine data collection has been achieved in recent years (Bushinsky et al., 2025), high-quality, ship-based pH measurements remain essential for ensuring the reliability of OA data collected by sensors on autonomous platforms, such as Argo floats and moorings, by enabling rigorous calibration protocols that correct for potential biases over time due to sensor drift, biofouling, and pressure effects (Maurer et al., 2021; Pérez et al., 2023; Takeshita et al., 2018).

Despite the critical role of pH data in understanding OA, significant limitations exist within key ocean databases such as the Global Ocean Data Analysis Project (GLODAPv2) (Olsen et al., 2016; Key et al., 2015). In GLODAPv2, pH data that is reported on the total hydrogen ion scale ($pH_T$)—which accounts for both the aqueous hydrogen ions ($H_3O^+$) and the associated form with sulfate ions ($HSO_4^-$)—and have historically been collected less frequently than other carbonate system variables, such as total alkalinity ($A_T$) and total dissolved inorganic carbon ($C_T$) (Key et al., 2015; Lauvset et al., 2024; Olsen et al., 2019). $A_T$ and $C_T$ measurements are generally considered more reliable due to the availability of standardized reference materials, consensually accepted methods, and quality control procedures. In contrast, although $pH_T$ measurements are technically precise, easy to perform, and cost-effective, their intercomparability is more challenging, arising from methodological inconsistencies across various research initiatives (Dickson et al., 2015; Ma et al., 2019; Álvarez et al., 2020; Capitaine et al., 2023). The lack of traceability to a common reference, preferentially the International System of units for spectrophotometric pH measurements (Dickson et al., 2015), and the unavailability of pH reference materials within the seawater pH range (Capitaine et al., 2023), together with the documented issues affecting $pH_T$ calculated from $A_T$ and $C_T$ (including $pH_T$-dependent offsets and larger propagated uncertainties; Álvarez et al., 2020; Carter et al., 2024b), mean that neither unadjusted direct observations nor calculated values currently provide a fully trusted global reference. Both limitations may therefore affect the reliability of pH data for climate-quality OA assessments.


Briefly, the spectrophotometric pH method is a straightforward technique that involves adding an acid-base
indicator dye to the seawater sample. The method was initially defined in the 1980s (Robert-Baldo et al., 1985;
Byrne and Breland 1989), and the use of meta-cresol purple (mCP) as the indicator dye began in the 1990s (Clayton
and Byrne, 1993). The technique has been updated since then, but it still lacks metrological traceability and
reference materials (Ma et al., 2019; Carter et al., 2024a). The method is based on the distinct absorbance
wavelengths of the indicator dye's acid and basic forms, which are used to calculate an absorbance ratio.
Subsequently, $pH_T$ is calculated using the indicator dye's dissociation constant and its extinction coefficients
through a parameterization in function of temperature and salinity, relating the absorbance ratio with $pH_T$. This
method offers a high degree of precision (error even lower than $\pm0.001$ pH units), with an approximate total
uncertainty of $\pm0.01$ $pH_T$ units (Dickson, 2010; Carter et al., 2024a). A primary source of error in these $pH_T$
measurements arises from the impurities present in the mCP dye itself (Liu et al., 2011). Over the last decade,
different studies proved that mCP impurities cause the measured $pH_T$ to exhibit a bias that is dependent on the
sample's $pH_T$ and the brand and batch of the mCP dye used (Liu et al., 2011; Yao et al., 2007). Consequently, mCP
was proposed to be purified (Liu et al., 2011; Rivaro et al., 2021) to remove these impurities and parameterizations
re-evaluated for those purified mCP dyes (Liu et al., 2011; Loucaides et al., 2017; Müller et al., 2018). Although
some laboratories, mostly in the US, currently use purified mCP dye (Carter et al., 2018), purified mCP dyes are
not commercially available, being scarce and expensive, and therefore not affordable for all laboratories.
Alternatively, it is possible to evaluate the effect of these impurities on the absorbance values and correct them
accordingly (Douglas and Byrne, 2017; hereafter DB'17; Takeshita et al., 2020, 2021; Woosley, 2021).

Over the past two decades, the GO-SHIP A25 OVIDE-BOCATS ship-based hydrographic section (OVIDE-
BOCATS hereafter; see Sect. 2.1)—the only transoceanic cruise with a biennial frequency in the CLIVAR (Climate
and Ocean: Variability, Predictability, and Change) and GO-SHIP programs—has built an extensive $pH_T$ time
series in the NA of more than 23,500 $pH_T$ samples measured using unpurified mCP dye. During the 11 OVIDE-
BOCATS cruises, $pH_T$ was measured spectrophotometrically following a consistent methodology and using the
same commercial mCP dye brand, Sigma-Aldrich, which—like other commercial brands—contains impurities (Liu
et al., 2011; Yao et al., 2007). As awareness of $pH_T$ biases introduced by mCP impurities has grown, so has the
need to assess and correct their impact to ensure the internal consistency and long-term comparability of the
OVIDE-BOCATS $pH_T$ dataset. In this context, we evaluated the bias induced by these impurities by carrying out
measurements with purified and unpurified mCP dyes, and assessed independent methods to account for and correct
the effect of these impurities in both the historical and contemporary OVIDE-BOCATS $pH_T$ data. Here we present
(1) the new data from the last two BOCATS cruises of 2021 and 2023, (2) the absorbance measurements along
with the evaluation of the impurities effect on them, and (3) the entire OVIDE-BOCATS $pH_T$ database product
since 2002, consistently adjusted for impurity-related bias. This effort allowed us to evaluate—in a consistent
way—the OA rates in the NA and to improve the reliability of the $pH_T$ data collected to date, which is fundamental
for understanding the ocean's response to climate change.

**2. Methods**
**2.1 OVIDE-BOCATS transoceanic section**
The OVIDE-BOCATS section is a high-quality hydrographic transect in the NA, extending from Portugal to
Greenland and largely following the GO-SHIP A25 track (Fig. 1), with the objective of studying the SPNA region.
Initially focused on physical oceanography (Mercier et al., 2024), its scope rapidly expanded to include critical
aspects of the carbon cycle, such as OA and the uptake and storage of $C_{ant}$ in the SPNA—one of the ocean's largest
$C_{ant}$ reservoirs (Sabine et al., 2004). Since 2002, in situ physical and on-board biogeochemical measurements have
been performed biennially along the OVIDE-BOCATS section. This repeated section is one of the longest-standing
and most frequently revisited transects within the GO-SHIP and CLIVAR programs. Accordingly, data
management follows strict policies, and all datasets are publicly available
(https://www.ncei.noaa.gov/access/ocean-carbon-acidification-data-
system/oceans/RepeatSections/clivar_ovide.html).
Thanks to its high frequency of repeated occupations, the OVIDE-BOCATS dataset offers a unique opportunity to
study the biennial evolution of NA processes. The program focuses on water mass properties, mixing, and
circulation; the impact of climate events on NA dynamics; volume and heat transports; and AMOC variability.
Particular attention is given to investigate key water masses such as Subpolar Mode Water (SPMW) and Labrador
Sea Water (LSW)—both formed through deep winter mixing in the SPNA— as well as Denmark Strait Overflow
Water (DSOW) and Iceland-Scotland Overflow Water (ISOW)—both resulting from the entrainment of SPMW
and LSW into the overflows at the sills between Greenland, Iceland, and Scotland, respectively (García-Ibáñez et
al., 2015; Lherminier et al., 2010).

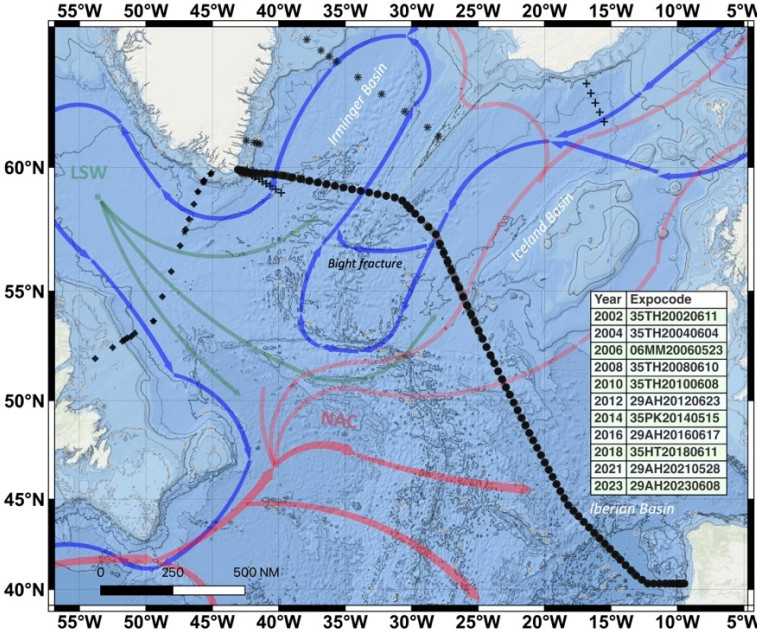

**Figure 1.** Bathymetric map showing the main water masses and circulation patterns within the SPNA region
covered by the OVIDE-BOCATS program (station locations indicated by black dots). Stations outside the main
OVIDE-BOCATS (A25) section are opportunistic stations, dependent on the ship's route, whose data are also
included in the final database product: "plus" symbols near Iceland and Greenland are stations sampled in 2006;
"pentagon" and "rhombus" located in the Labrador Sea are stations sampled during 2012 and 2014 cruises; and
"asterisk" symbols are stations sampled in 2023, near Greenland and in the Irminger Sea. The inset table lists each
biannual cruise along with its corresponding expocode as identified in GLODAP. Major currents and water masses
are illustrated in different colors according to their temperature and depth (from red—warmer and shallower—to
blue—colder and deeper). The main schematized currents and water masses include the North Atlantic Current
(NAC), Labrador Sea Water (LSW), and Iceland-Scotland Overflow Water (ISOW). The principal basins traversed
along the OVIDE-BOCATS section, from west to east, are the *Irminger*, *Iceland*, and *Iberian basins*.
Research within the OVIDE-BOCATS framework has also focused on carbon inventories and the deep
biogeochemical imprint, such as the role of the Deep Western Boundary Current (DWBC) in transporting oxygen,
nutrients, and dissolved organic carbon (Álvarez-Salgado et al., 2013; Fontela et al., 2019; 2020b). In addition,
OVIDE-BOCATS has evaluated OA rates in the NA (Fontela et al., 2020a; García-Ibáñez et al., 2016; Vázquez-
Rodríguez et al., 2012) and analyzed its impact on marine biodiversity (Pérez et al., 2018; García-Ibáñez et al.,
2021) as well as the ocean's capacity to absorb, store, and transport $CO_2$ (Bajon et al., 2025, manuscript in review;
Pérez et al., 2013; Zunino et al., 2015). In summary, the sustained observations from the OVIDE-BOCATS program
demonstrated to be essential for detecting OA trends, improving climate models, and understanding the SPNA's
response to climate change (DeVries et al., 2023; Rodgers et al., 2023).
**2.2 $pH_T$ determination**
*2.2.1 Spectrophotometric $pH_T$ method fundamentals*
In all OVIDE-BOCATS cruises, $pH_T$ was measured manually following the spectrophotometric method proposed
by Clayton and Byrne (1993)—hereafter CB'93. This method involves adding a mCP dye solution to the seawater
sample and calculating the sample's $pH_T$ using the following equation:
$$pH_T = pK_2 + \log_{10}([I^{-2}] / [HI^-]) \tag{1},$$
where $[HI^-]$ and $[I_2^-]$ represent the concentrations of mono-dissociated and bi-dissociated species of the indicator
dye, respectively. The concentration ratio ($[I^{-2}] / [HI^-]$) can be determined spectrophotometrically by measuring
absorbance at the corresponding maximum absorbance wavelengths (A), i.e., 434 nm and 578 nm, respectively,
corrected for baseline absorbance at 730 nm—hereafter referred to as $_{434}A$, $_{578}A$, and $_{730}A$, respectively. $pH_T$ is then
calculated using CB'93 parameterization of Eq. (1):
$$pH_T = 1245.69/T + 3.8275 + (2.11 \cdot 10^{-3}) \cdot (35 - S) + \log((R - 0.0069) / (2.222 - 0.133 \cdot R)) \tag{2},$$
where $T$ is temperature in Kelvin, $S$ is salinity, and $R$ is the ratio of the absorbances of the mono-dissociated and
bi-dissociated forms of the indicator dye corrected for baseline absorbance:
$$R = (_{578}A - _{730}A) / (_{434}A - _{730}A) \tag{3}.$$
The first three terms of Eq. (2) represent the second dissociation constant of mCP dye ($pK_2$, or $-pK_1$ in CB'93). The
$-pK_1$ obtained by CB'93 is based on the *TRIS* buffer characterization of Dickson (1993), which used electromotive
force data from Ramette et al. (1977; Lee et al., 2000). DelValls and Dickson (1998)—hereafter DVD'98—later
determined that the $pH_T$ values assigned to *TRIS* buffers needed to be increased by 0.0047, for all temperatures and
salinities. These corrected *TRIS* $pH_T$ values have recently been confirmed by Müller et al. (2018). Consequently,
spectrophotometric $pH_T$ values obtained using the CB'93 parameterization should be adjusted by +0.0047 $pH_T$ units
(DVD'98; Lee et al., 2000).
The CB'93 parameterization was developed using Kodak mCP dye, prepared in deionized water, which contained
impurities contributing significant absorbance at 434 nm (referred to as $_{434}A_{imp}$). However, it was not until the 2000s
that the impact of impurities on $pH_T$ measurements was evidenced. Specifically, Yao et al. (2007) compared $pH_T$

determinations using Sigma-Aldrich and Kodak mCP dyes with *TRIS* buffers, finding that $pH_T$ values obtained with Sigma-Aldrich mCP dye were between 0.001 to 0.006 $pH_T$ units higher than those with Kodak mCP dye (for $pH_T$ ranging from 7.4 to 8.2), attributed to lower $_{434}A_{imp}$ values in the Sigma-Aldrich mCP dye. Later, Liu et al. (2011; hereafter L'11) purified mCP dye and developed a new parameterization to determine $pH_T$ from R, demonstrating that applying their new parameterization to R data measured with impure mCP dye results in $pH_T$ values up to 0.018 $pH_T$ units lower. Subsequently, Loucaides et al. (2017) and Müller et al. (2018) produced very similar parameterizations, extending the valid salinity and temperature ranges and confirming the same $pH_T$ versus R relationship at 25ºC and oceanic salinities (see Fig. S1 in Álvarez et al., 2025. Thus, ideally, $pH_T$ measurements should be performed using purified, well-characterized mCP dyes and following a consensus method that ensures traceability to the International System of Units (SI; Capitaine et al., 2023). However, the purification procedure is not accessible to many laboratories routinely measuring seawater $pH_T$. To overcome this limitation and facilitate high-quality spectrophotometric $pH_T$ measurements, DB'17 proposed a method to determine $_{434}A_{imp}$ and an associated correction procedure. This approach allows R to be corrected for the contribution of impurities at $_{434}A$ (i.e., $_{434}A_{imp}$), and consequently enables $pH_T$ calculations using parameterizations derived for purified mCP dye, such as the L'11 parameterization.

What is the bias introduced in $pH_T$ measurements as a result of mCP dye impurities? Most of the spectrophotometric $pH_T$ values in GLODAPv2 (Lauvset et al., 2024) are calculated using the CB'93 parameterization, based on measurements made with mCP dyes that contain impurities. Figure 2 shows a family of curves representing the theoretical differences between $pH_T$ values calculated using the CB'93 parameterization with R values that would have been obtained with unpurified mCP dyes (i.e., with varying $_{434}A_{imp}$), and those obtained using the L'11 parameterization applied to R values corresponding to a fully purified mCP dye (i.e., $_{434}A_{imp} = 0$). We computed the corresponding $_{434}A$ values ($_{434}A_{pur}$, i.e., $_{434}A_{imp} = 0$) using the relationship described in Sect. 2.2.4 for a set of theoretical purified R ($R_{pur}$) values ranging from 0.3 to 2.6. Both the purified R values and their associated $_{434}A$ values were then used in Eq. (11) of DB'17 to compute the adjusted R values ($R_{unpur}$; referred to as $R_{obs}$ in DB'17) that reflect the contribution of mCP dye impurities ($_{434}A_{imp} \neq 0$) as follows:

$$R_{unpur} = R_{pur} / (1 + (_{434}A_{imp} /_{434}A_{pur}) \qquad (4).$$

The mCP dye brands and their corresponding absorbance values due to their impurities, as shown in Fig. 2, are primarily taken from Table 2 of DB'17, that are based on an mCP dye concentration of 3.3 µM in the sample cell. When the CB'93 parameterization is applied to $R_{pur}$ ($_{434}A_{imp} = 0$) at S = 35 and 25ºC, the largest theoretical $pH_T$ differences (> 0.015 $pH_T$ units) are observed relative to $pH_T$ values obtained by applying the L'11 parametrization to the corresponding $R_{pur}$ under the same conditions (see turquoise line in Fig. 2). In contrast, when CB'93 parametrization is applied to $R_{unpur}$ values obtained with Kodak mCP dye, the resulting $pH_T$ values differ by only $\pm0.003$ $pH_T$ units from those obtained with the L'11 parameterization with a purified mCP dye, with minimal differences observed in the $pH_T$ range of 7.65 to 8.15 (purple line in Fig. 2). This agreement arises because the CB'93 parametrization—developed using Kodak mCP dye and calibrated against *TRIS* buffers using unpurified mCP dye—yields lower R values than that obtained with purified mCP dye. As a result, both parameterizations converge around the *TRIS* buffer $pH_T$ value (8.093 at 25ºC; DVD'98). Indeed, Fig. 2 shows that larger biases in the final $pH_T$ arise when using the CB'93 parameterization with mCP dyes with lower impurity content, while the magnitude and $pH_T$-dependence of these biases increases with higher impurity levels.

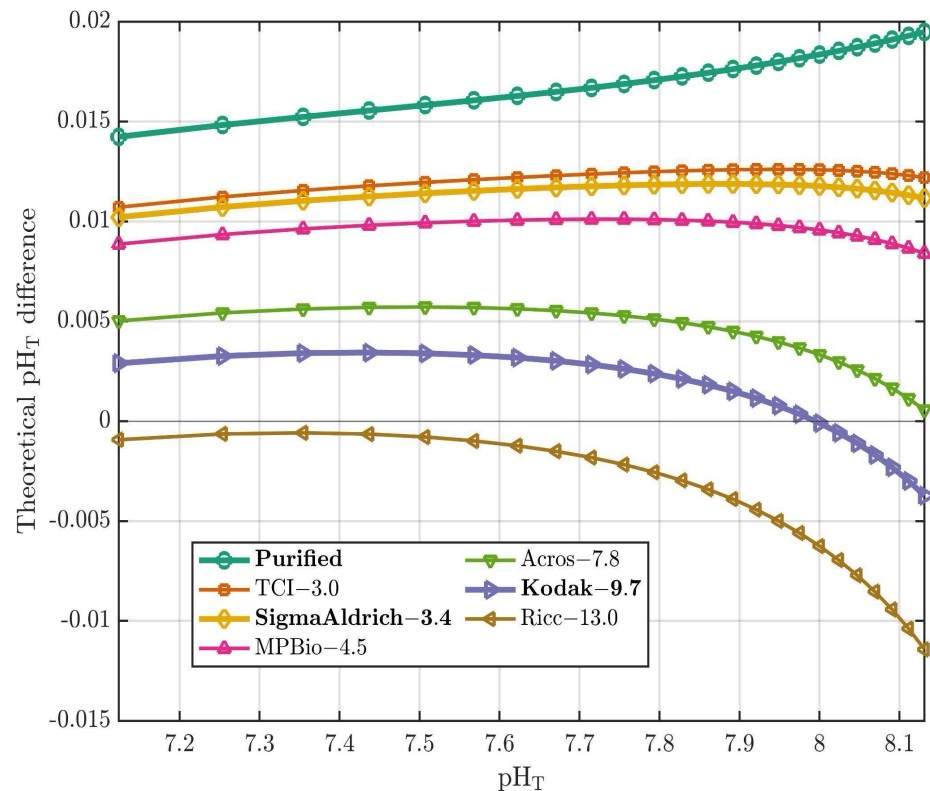

**Figure 2.** Theoretical differences in $pH_T$ (y-axis) between values using the CB'93 parametrization with the DVD'98
correction (applied to R values derived from mCP dyes with varying impurities) and those calculated using the
L'11 parametrization (applied to R values derived from a purified mCP dye). The differences are plotted against
$pH_T$ computed using the L'11 parameterization (x-axis), at S = 35 and T = 25ºC. Each mCP dye is represented by
a different color, with its corresponding $_{434}A_{imp}$ value (in units of $10^{-3}$ absorbance) specific to the lot used in DB'17,
listed after the indicator dye name. Bold indicates mCP dye brands discussed in this work. All $_{434}A_{imp}$ values are
taken from DB'17 and are specific to the lot used, except for Sigma-Aldrich, whose value was determined in this
study.
*2.2.2 $pH_T$ sampling and manual spectrophotometric procedure during the OVIDE-BOCATS cruises*
On all OVIDE-BOCATS cruises, seawater samples for $pH_T$ were collected after oxygen samples, using 100 mm
pathlength cylindrical optical glass cells with two stoppers (Dickson et al., 2007). Each sample was taken by rinsing
the optical cell twice and flushing it with seawater two or three times its volume, ensuring no air bubbles remained.
To achieve this, the cell was held with the outlet above the inlet, the outlet was plugged first, the sampling tube
was removed, and the second plug was inserted—taking care to avoid air bubbles. After rinsing and externally
drying the cells, they were placed in a thermostatic incubator set at 25°C for at least 30—45 minutes to ensure
temperature stabilization at 25ºC prior to their measurement.
For each sample, a blank measurement was performed after drying and cleaning both faces of the optical cell and
placing it in the spectrophotometer's cell holder. Following blanking at the three target wavelengths (434 nm, 578
nm, and 730 nm) with sampled seawater, 75 µL of 2 mM mCP dye solution were added to each 28 mL sample cell
using an adjustable repeater pipette (SOCOREX), resulting in a final mCP dye concentration of 5.36 µM in the

cell. Dispenser syringes were wrapped in aluminum foil to prevent photodegradation of the mCP dye (Fontela et al., 2023). After the mCP dye addition, the cell was thoroughly shaken and placed back in the holder in the same orientation as for the blanking, and triplicate absorbance readings were carried out at the same target wavelengths as for the blank. All absorbance readings were carried out in the spectrophotometer's thermostatted cell compartment, maintained at $25.0 \pm 0.2$°C.

Byrne and Breland (1989) demonstrated that R measurements are largely insensitive to small temperature variations, for cresol red dye. The same general behaviour applies to mCP dye, in which R values between 0.5 and 2.8 exhibit $pH_T$ errors of less than 0.001 $pH_T$ units per 1°C change when using the L'11 parameterization. This insensitivity arises because, for mCP, the temperature dependence of the indicator dye's $pK_2$ closely parallels the temperature dependence of seawater pH. The CB'93 parametrization shows a slightly greater temperature sensitivity, such that temperature deviations must be kept within approximately $\pm 0.5$°C to limit the $pH_T$ error to $\leq$ 0.0012 $pH_T$ units. Therefore, it is recommended to ensure that temperature deviations remain within $\pm 0.5$°C of the reference temperature (25°C). In our procedure, sample temperature was monitored every five measurements to verify that it remained within this tolerance.

### 2.2.3 Effect of the indicator dye addition and spectrophotometer performance on the $pH_T$ measurements

Throughout the OVIDE-BOCATS program, the mCP dye used was from Sigma-Aldrich (Cat. No. 11,436-7 in the basic form; $C_{21}H_{17}NaO_5S$; molecular weight 404.41 g), with only a few different batches from this brand used over the 11 cruises. Prior to 2018, mCP dye solutions were prepared by dissolving 0.080 g of the mCP sodium salt in 100 mL of natural seawater. Following DB'17 recommendations, the preparation method was modified in 2018, with the mCP dye being dissolved in 0.7 M NaCl instead of seawater. The absorbances of the mCP dye solutions at 434 nm, 578 nm, and 730 nm were measured spectrophotometrically using a 0.1 mm optical cell to ensure that the R values remained close to 1, corresponding to a $pH_T$ of approximately 7.67 (Li et al., 2020; at S = 35 and T = 25°C). mCP dye solutions were stored in Pyrex bottles, refrigerated, and protected from light using aluminum foil.

The addition of the indicator dye slightly perturbs the sample's $pH_T$, with the magnitude of this effect increasing for shorter optical pathlengths and when the $pH_T$ difference between the sample and the indicator dye is large (Chierici et al., 1999). For instance, when using a 100 mm optical pathlength, the indicator dye induced $pH_T$ perturbation is typically less than 0.006 $pH_T$ units within the $pH_T$ range of 7.6 to 8.0 (Chierici et al., 1999; Li et al., 2020; Table 1), which is relatively minor but not negligible. A common approach to account for this effect, as proposed by CB'93, involves performing a double addition of the indicator dye solution to the samples and calculating the difference in the resulting R values ($\Delta R$) between the first (single) and second (double) addition. This $\Delta R$ correction is then applied to the measured R values. Alternatively, the correction can be expressed in terms of $\Delta pH_T$, which may be directly applied to the computed $pH_T$ values (Takeshita et al., 2020, 2022). Both $\Delta R$ and $\Delta pH_T$ correction approaches were evaluated. During each cruise, between two and four double indicator dye addition experiments were performed. In each experiment, seawater samples were modified to obtain four $pH_T$ values ranging from 7.4 to 8.2, with four samples per $pH_T$ level (N = 16 samples per experiment). Following blanking, an initial addition of 50 µL of mCP dye solution was made to each sample, and absorbance was measured as described in Section *2.2.2*. A second addition of 50 µL of mCP dye solution was then made (resulting in a total of 100 µL of mCP dye solution added), and absorbance measurements were repeated. These double-addition experiments enabled the determination of linear regressions of the change in $pH_T$ ($\Delta pH_T = pH_{T,2} - pH_{T,1}$; where subscripts *2* and *1* refer to 100 µL and 50 µL of mCP dye solution added, respectively) or R ($\Delta R = R_2 - R_1$) as a function of the initial $pH_{T,1}$ or $R_1$, respectively (Fig. S1 in the Supplement). This two-step 50 µL addition bracketed

the typical 75 µL reference volume added to the sample, allowing us to evaluate the dye effect on the $\Delta R$ or $\Delta pH_T$ both below and above this reference. The corresponding relationship was expressed as:

$$\Delta pH_T = a \cdot (pH_T - pH_T^{y=0}) \tag{5},$$

where $a$ is the slope of the linear regression and $pH_T^{y=0}$ represents the $pH_T$ at which the indicator dye addition has no effect. An analogous expression was used for $\Delta R$ [$\Delta R = a' \cdot (R - R^{y=0})$, being $a'$ the particular slope for the $\Delta R$-R linear regression]. Since the standard volume of mCP dye solution used in OVIDE-BOCATS cruises was 75 µL, while the double addition experiments used 50 µL additions, a correction factor of 75/50 was applied to adjust both $\Delta pH_T$ and $\Delta R$. The corrected $pH_T$ ($pH_{T,corrected}$) was thus calculated as:

$$pH_{T,corrected} = pH_m - 75/50 \cdot a \cdot (pH_m - pH_T^{y=0}) \tag{6},$$

where $pH_m$ is the uncorrected measured $pH_T$ (i.e., prior to its $\Delta pH_T$ correction) [analogously: $R_{corrected} = R - 75/50 \cdot a' \cdot (R_m - R^{y=0})$]. At $pH_T^{y=0}$, the R ($pH_T$) of the original sample and the R ($pH_T$) of the indicator dye are the same, so no change is observed. If $pH_m > pH_T^{y=0}$, then $pH_{T,2}$ (or $R_2$) < $pH_{T,1}$ (or $R_1$), as the mCP dye addition lowers $pH_m$; hence, $pH_{T,corrected}$ will be higher than $pH_m$. Conversely, if $pH_m < pH_T^{y=0}$, the mCP dye addition increases the $pH_m$, and $pH_{T,corrected}$ will be lower than $pH_m$.

The linear regressions of $\Delta R$ versus $R_1$ ($\Delta R$-vs-$R_1$) and $\Delta pH_T$ versus $pH_{T,1}$ ($\Delta pH_T$-vs-$pH_{T,1}$) obtained for each OVIDE-BOCATS cruise are summarized in Table 1. The slopes for $\Delta R$-vs-$R_1$ range from 0.0048 (BOCATS2-2023 cruise) to 0.0230 (OVIDE-2006 cruise), with the perturbation vanishing ($\Delta R = 0$) when R≈1.0 ± 0.2, i.e., when the sample $pH_T$ closely matches that of the mCP dye solution. Carter et al. (2013) proposed a methodological refinement by normalizing $\Delta R$ with the change in absorbances at the isosbestic point ($\Delta_{488}A$; see Section *2.2.4*), improving the robustness of the correction. Accordingly, using the Carter et al. (2013) approach [$\Delta(R/_{488}A)$ versus $R_1$; $\Delta(R/_{488}A)$-vs-$R_1$] increased the overall explained variability ($R^2$) of the linear fits. Similarly, for $\Delta pH_T$-vs-$pH_{T,1}$ regressions, the smallest slope was recorded for BOCATS2-2021 cruise and the largest again in the OVIDE-2006 cruise (Table 1). When using $\Delta(pH_T/_{488}A)$-vs-$pH_{T,1}$, the distribution of regression slopes was similar but less variable (average regression: -0.0431 ± 0.019·($pH_T$ - 7.73 ± 0.09); Table 1), consistent with the mCP dye perturbation trends reported by Takeshita et al. (2022) [$\Delta pH_T/\Delta_{488}A$ = -0.042 ± 0.003·($pH_T$ - 7.76); N = 91]. The OVIDE-BOCATS $pH_T$ data were corrected using the cruise-specific $\Delta pH_T/\Delta_{488}A$ relationships. Overall, the evaluation of the mCP dye's perturbation on the sample's $pH_T$ was consistent across all cruises, regardless of whether $\Delta R$- or $\Delta pH_T$-based methods were used.

In addition to the impact of the mCP dye addition, both the $\Delta R$ and $\Delta pH_T$ corrections can be influenced by the performance characteristics of the spectrophotometer used (Carter et al., 2013; Álvarez et al., 2020; Takeshita et al., 2021; Fong et al., 2024). If the spectrophotometer follows the Beer-Lambert law (i.e., no optical non-linearity), the effect of the mCP dye addition results in a linear relationship for both $\Delta R$-vs-$R_1$ and $\Delta pH_T$-vs-$pH_{T,1}$ regressions (Li et al., 2020), meaning that the relationships are only affected by chemistry. This only-chemical effect was evaluated by Li et al. (2020) over a wide range of salinities and $A_T$s, including those present during OVIDE-BOCATS cruises, allowing us to replicate their chemical model (blue diamonds in Fig. 3). The $\Delta R$-vs-$R_1$ regression exhibits better linearity than $\Delta pH_T$-vs-$pH_{T,1}$ regression, though both reflect the mCP dye's interference in the physico-chemical ionic equilibrium of the marine carbonate system. However, when introducing a 0.04% deviation from linearity in the spectrophotometer (i.e., loss of the Beer-Lambert behavior, thus including an additional impact to the chemical effect) the regression slopes for both $\Delta pH_T$-vs-$pH_{T,1}$ and $\Delta R$-vs-$R_1$ roughly double, revealing that

this instrumental non-linearity amplifies the chemistry effect of the mCP dye. Notably, while $\Delta pH_T$ remains linear
during this distorsion, $\Delta R$ becomes non-linear (orange squares in Fig. 3). When R = 1 (i.e., $_{434}A = _{578}A$ and sample
$pH_T$ = mCP dye $pH_T$), non-linearity impacts both absorbances equally, resulting in no change in R and therefore in
$pH_T$. In contrast, deviations from R = 1 (or $pH_T$ = 7.65; at S= 35 and T = 25ºC) enhance this artifact due to
increasingly unequal absorbances at $_{434}A$ and $_{578}A$, leading to spurious R values and biased $pH_T$.
**Table 1**. Summary of each OVIDE-BOCATS cruise alias, the spectrophotometer used, and the mean and standard
deviation of $_{488}A$ values (from 2002 to 2018 estimated using Eq. (7), from 2018 to 2023 directly measured). The
table also includes the linear regression equations and their explained variance ($R^2$) for the $\Delta R$-vs-$R_1$, $\Delta(R/_{488}A)$-
vs-$R_1$, $\Delta pH_T$-vs-$pH_{T,1}$, and $\Delta(pH_T/_{488}A)$-vs-$pH_{T,1}$ relationships. Additionally, it reports the corresponding $\Delta pH_T$ at
$pH_T$ = 7.7 and $pH_T$ = 8.0, calculated using the $\Delta(pH_T/_{488}A)$-vs-$pH_{T,1}$ relationship for each cruise. Note that $\Delta R$-vs-
$R_1$ and $\Delta pH_T$-vs-$pH_{T,1}$ regressions are based on an addition of 50 µL mCP dye solution; therefore, a correction
factor (e.g., 75/50) must be applied when using $\Delta R$-vs-$R_1$ and $\Delta pH_T$-vs-$pH_{T,1}$ relationships to samples measured
with a different addition volumes (e.g., 75 µL).

| CRUISE | Spectrophotometer | $_{488}A$ | $\Delta R$-vs-$R_1$ \|\| $R^2$ | $\Delta(R/_{488}A)$-vs-$R_1$ \|\| $R^2$ | $\Delta pH_T$-vs-$pH_{T,1}$ \|\| $R^2$ | $\Delta(pH_T/_{488}A)$-vs-$pH_{T,1}$ \|\| $R^2$ | $\Delta pH_T$ at~7.7 | $\Delta pH_T$ at~8.0 |
|---|---|---|---|---|---|---|---|---|
| Ovide-2002 | CECIL-3041 | 0.331±0.020 | -0.0068·(R-0.82)\|\| 0.32 | -0.059·(R-1.07) \|\|0.41 | -0.0078·(pH$_T$-7.71) \|\|0.51 | -0.056·(pH$_T$-7.71) \|\|0.59 | 0,000 | 0,005 |
| Ovide-2004 | Shimadzu UV-2401PC | 0.232±0.020 | -0.0092·(R-0.96) \|\| 0.78 | -0.038·(R-0.97) \|\|0.81 | -0.0078·(pH$_T$-7.70) \|\|0.70 | -0.033·(pH$_T$-7.70) \|\|0.73 | 0,000 | 0,002 |
| Ovide-2006 | Shimadzu UV-2401PC | 0.250±0.023 | -0.035·(R-1.10) \|\| 0.92 | -0.134·(R-1.00) \|\|0.94 | -0.023· (pH$_T$-7.70) \|\|0.88 | -0.103· (pH$_T$-7.68) \|\|0.96 | 0,001 | 0,008 |
| Ovide-2008 | Shimadzu UV-2401PC | 0.230±0.015 | -0.0060·(R-1.2) \|\| 0.60 | -0.029·(R-1.04) \|\|0.84 | -0.0084·(pH$_T$-7.83) \|\|0.89 | -0.037·(pH$_T$-7.84) \|\|0.89 | -0,001 | 0,001 |
| Ovide-2010 | Shimadzu UV-2401PC | 0.233±0.015 | -0.022·(R-1.06) \|\| 0.78 | -0.096·(R-1.06) \|\|0.81 | -0.0106·(pH$_T$-7.68) \|\|0.70 | -0.069·(pH$_T$-7.68) \|\|0.73 | 0,000 | 0,005 |
| CATARINA-2012 | Perkin Elmer Lambda 800 UV-VIS | 0.211±0.016 | -0.014·(R -0.80) \|\| 0.93 | -0.100·(R -0.81) \|\|0.94 | -0.0090·(pH$_T$-7.55) \|\|0.89 | -0.065·(pH$_T$ -7.58) \|\|0.93 | 0,002 | 0,006 |
| Geovide-2014 | Shimadzu UV-2401PC | 0.218±0.007 | -0.0066·(R -1.12) \|\| 0.76 | -0.046·(R -1.16) \|\|0.75 | -0.0082·(pH$_T$-7.77) \|\|0.76 | -0.065·(pH$_T$ -7.77) \|\|0.71 | -0,001 | 0,003 |
| BOCATS-2016 | Perkin Elmer Lambda 800 UV-VIS | 0.369±0.022 | -0.0070·(R-1.19) \|\| 0.81 | -0.028·(R-1.19) \|\|0.81 | -0.0080·(pH$_T$-7.85) \|\|0.91 | -0.033·(pH$_T$ -7.85) \|\|0.91 | -0,002 | 0,002 |
| Ovide-2018 | Shimadzu UV-2401PC | 0.358±0.027 | -0.0108·(R-1.09) \|\| 0.72 | -0.046·(R-1.04) \|\|0.77 | -0.0083·(pH$_T$- 7.73) \|\|0.69 | -0.030·(pH$_T$ -7.76) \|\|0.70 | -0,001 | 0,003 |
| BOCATS2-2021 | Perkin Elmer Lambda 800 UV-VIS | 0.359±0.027 | -0.0070·(R-1.16) \|\| 0.96 | -0.027·(R-1.16) \|\|0.96 | -0.0050·(pH$_T$- 7.76) \|\|0.89 | -0.021·(pH$_T$ - 7.76) \|\|0.90 | 0,000 | 0,002 |
| BOCATS2-2023 | Perkin Elmer Lambda 800 UV-VIS | 0.389±0.032 | -0.0048·(R-1.12) \|\| 0.92 | -0.020·(R-1.12) \|\|0.92 | -0.0063·(pH$_T$- 7.80) \|\|0.95 | -0.023·(pH$_T$ - 7.76) \|\|0.96 | -0,001 | 0,002 |

These results suggest that the $\Delta pH_T$-vs-$pH_{T,1}$ relationship provides a more accurate assessment of the mCP dye's
effect on the sample's $pH_T$ when the spectrophotometer exhibits even slight deviations from the Beer-Lambert law.
Indeed, the small differences between the slopes of the $\Delta R$-vs-$R_1$ and $\Delta pH_T$-vs-$pH_{T,1}$ regressions reported in Table
1 can be attributed to the distinct ways in which $pH_T$ and R respond to such nonlinearity. The theoretical slopes of
$\Delta pH_T$-vs-$pH_{T,1}$ shown in Fig. 3 are consistent with those derived experimentally (Table 1). The steepest
experimental slopes observed during the cruises may reflect greater deviations from the Beer-Lambert law, which
can depend on both the spectrophotometer used and the $pH_T$ range of the seawater batches used in these
assessments. Conversely, cruises with $\Delta pH_T$-vs-$pH_{T,1}$ slopes closer to the chemical model prediction—such as in
the 2021 and 2023 cruises—indicate better spectrophotometer performance. Nevertheless, the differences in
$pH_{T,corrected}$ when applying either $\Delta R$-vs-$R_1$ or $\Delta pH_T$-vs-$pH_{T,1}$ corrections remain small (< 0.001 $pH_T$ units; see Fig.
S2 in the Supplement), implying that the choice of correction method has minimal impact on the final estimation
of the mCP dye perturbation. For consistency, OVIDE-BOCATS $pH_T$ data were corrected using the $\Delta pH_T/\Delta_{488}A$
approach specific to each cruise.

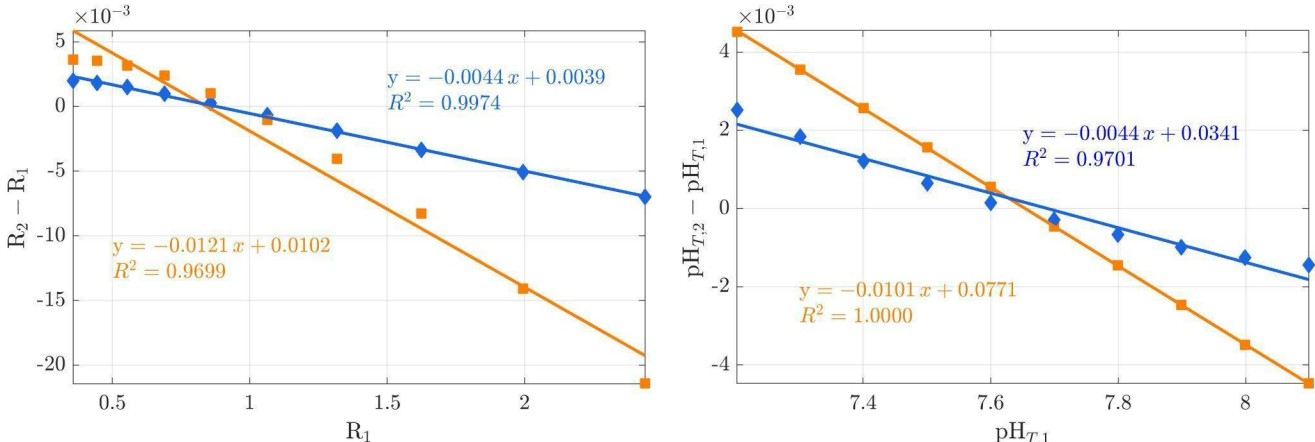

**Figure 3**. Theoretical evaluation of the difference of the impact of mCP dye perturbation on the sample's $pH_T$ (S = 35; T = 25°C; $A_T$ = 2300 µmol kg$^{-1}$), depending on whether the spectrophotometer behaves linearly—i.e., follows the Beer-Lambert law—or not. The left and right panels depict the $\Delta R$-vs-$R_1$ and $\Delta pH_T$-vs-$pH_{T,1}$ relationships, respectively. Blue diamonds represent an ideal, linear spectrophotometer response, derived from the chemical model of Li et al. (2020) based on single and double additions of 50 µL of mCP dye solution ([2 mM]) to a 28 mL sample cell with a 10 cm pathlength ($_{488}A$ = 0.242), assuming a $pH_T$ of the mCP dye solution of 7.65). Orange squares incorporate an attenuation factor of 0.04% to simulate a deviation from linearity, thus representing the combined effect of spectrophotometer non-linearity and perturbation in the physico-chemical ionic equilibrium.

### *2.2.4 Absorbance at the isosbestic point*

During the OVIDE-BOCATS cruises, mCP dye was manually added to samples using an adjustable repeater pipette (see Section *2.2.2)*. For each cruise, volume deviations associated with manual addition were assessed by comparing each sample to the cruise-specific mean $_{488}A$, computed from all field-layer samples (~2000 per cruise). Manual addition resulted in volume deviations exceeding 20% in approximately 3% of the samples (~ 706 cases), potentially affecting $\Delta R$ and $\Delta pH_T$ determinations. To address variability in the volume of mCP dye solution added, Carter et al. (2013) recommended measuring absorbance at the isosbestic point ($_{488}A$), which provides a reliable proxy for actual mCP dye concentration in the sample cell. Accurate quantification of the mCP dye concentration is particularly important when applying the DB'17 methodology for impurity correction, since the $_{434}A_{imp}$ value is directly proportional to the mCP dye content. Therefore, sample-specific estimates of mCP dye concentration via $_{488}A$ allow for a more precise estimate of the $_{434}A_{imp}$ value.

Since 2018, OVIDE-BOCATS cruises have incorporated measurements at $_{488}A$ following the recommendation by Carter et al. (2013). The additions of 75 µL of 2 mM mCP dye solution to 28 mL seawater sample resulted in averaged $_{488}A$ values of 0.359 ± 0.027 (N = 2,193), 0.358 ± 0.027 (N = 2,154), and 0.377 ± 0.032 (N = 2,342) during the 2018, 2021, and 2023 cruises, respectively. These averages were not statistically different from one another, resulting in a $_{488}A$ mean value of 0.370 ± 0.033 for the period 2018—2023. This $_{488}A$ mean value was used to derive a parameterization for estimating $_{488}A$ from R values, fitted for an mCP dye concentration in the cell of 5.36 µM—particularly useful for pre-2018 cruises, where $_{488}A$ was not measured. For these earlier cruises, $_{488}A$ was estimated using the following parameterization:

$$_{488}A = {}_{578}A \cdot (-2.5486\ R^{2.5} + 17.338\ R^2 - 46.779\ R^{1.5} + 63.109\ R - 43.393\ R^{0.5} + 12.962) \tag{7}$$

This fit, based on 6,910 samples from the 2018, 2021, and 2023 cruises (Fig. S3 in the Supplement), explained
98.9% of the variance ($R^2 = 0.989$) and reproduces $_{488}A$ with a mean error of 0.002 ± 0.010. It enabled a more
accurate estimation of mCP dye concentrations across all cruises and improved the assessment of mCP dye impurity
effects on $_{434}A$ (see Sect. 3.4). Additionally, it enhanced the accuracy of mCP dye perturbation corrections using
$\Delta R/_{488}A$ and $\Delta pH_T/_{488}A$ (Table 1). A simplified parameterization using $_{434}A$ (e.g., $_{434}A = -0.0747 \cdot R + 0.403$ for 3.3
µM mCP dye concentration in the sample cell) is possible, but requires correction for $_{434}A_{imp}$.

### 2.2.5 $pH_T$ measurement repeatability

Throughout the 11 OVIDE-BOCATS cruises, a total of 502 duplicate samples were collected to evaluate the
reproducibility of $pH_T$ measurements using an unpurified mCP dye. At selected stations, two Niskin bottles were
closed at the same depth to obtain replicates. Any uncertainty introduced by collecting duplicates on two different
Niskin bottles (e.g., small leaks, biological activity, or delay in closing) was neglected. Figure 4 displays the
absolute $pH_T$ differences between replicates for each cruise. The overall mean and standard deviation of these
differences is 0.0014 ± 0.0015 $pH_T$ units (N = 502). The highest reproducibility was obtained in 2021, with 92
duplicate samples yielding a mean difference of 0.0007 ± 0.0010 $pH_T$ units. This improved reproducibility,
particularly evident during the two most recent cruises, coincides with more precise evaluations of mCP dye effects
(Table 1) and likely reflects the better performance of the spectrophotometer used (PerkinElmer Lambda 800; Table
1), contributing to the overall improvement in data quality. Typical reproducibility across OVIDE-BOCATS
cruises ranged between 0.0007 and 0.0018 $pH_T$ units.

## 3. Assessment of the effect of indicator dye impurities on $pH_T$

During the OVIDE-2018, BOCATS2-2021, and BOCATS2-2023 cruises, paired measurements were performed in
duplicate samples collected from the same Niskin bottle and measured using two types of mCP dye: (i) purified
mCP, provided by Dr. Byrne's laboratory, University of South Florida, USA (FB6 batch), and (ii) unpurified mCP,
commercially available from Sigma-Aldrich (Cat. No. 211761-5G, batch #07005HH). $pH_T$ values were obtained
applying the L'11 parametrization to R values obtained with purified mCP dye, and the CB'93 parametrization
combined with the DVD'98 correction (CB'93+DVD'98 hereafter) to R values obtained with unpurified mCP dye.

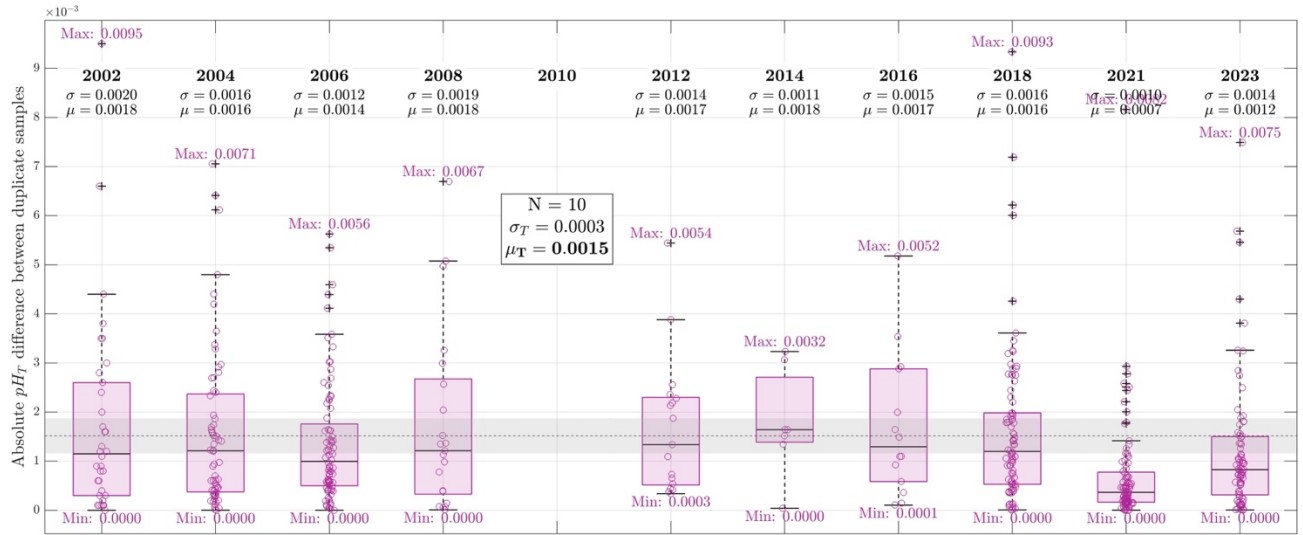


**Figure 4**. Whisker boxplots showing $pH_T$ repeatability across OVIDE-BOCATS cruises (2002—2023), represented as the absolute difference between duplicate samples collected during each cruise. The number of duplicate samples analyzed per cruise were: 34 (2002), 64 (2004), 84 (2006), 21 (2008), 0 (2010), 17 (2012), 7 (2014), 14 (2016), 90 (2018), 92 (2021), and 79 (2023). For each cruise, the subset mean ($\mu$) and standard deviation ($\sigma$) are indicated, as well as the minimum and maximum values. The overall mean ($\mu_T$) and standard deviation ($\sigma_T$) across all 10 cruises are shown in the inset textbox. $\mu_T$ is also plotted as a horizontal dashed line, with $\sigma_T$ represented as a shaded gray band. Each boxplot displays the median (horizontal line), first (Q1) and third (Q3) quartiles (box edges), and the minimum and maximum within 1.5 times the interquartile range (IQR = Q3 – Q1) (whiskers).

### 3.1 *TRIS* buffer validation

*TRIS* buffers in synthetic seawater (batch numbers 30 and 40) were obtained from Prof. Dickson's laboratory (Scripps Institution of Oceanography; USA). These buffers were supplied in 125 mL borosilicate glass bottles and contained an equimolar mixture of *TRIS/TRIS*-HCl in a synthetic seawater of nominal salinity 35. The reference $pH_T$ values of these batches can be calculated following DVD'98. Multiple bottles from both batches were measured during the BOCATS2-2021 and BOCATS2-2023 cruises using two mCP dye solutions: (i) unpurified mCP (75 µL of 2 mM solution; Sigma-Aldrich; 5.36 µM final mCP dye concentration in the sample cell), and (ii) purified mCP (10 µL of 11 mM solution; provided by Prof. R. Byrne's laboratory; 3.93 µM final mCP dye concentration in the sample cell).

Each buffer sample was measured in quadruplicate. Following blank measurement, the sample was placed in the spectrophotometer's thermostated cell holder and allowed to equilibrate for 10—15 minutes. Measurements were then conducted at 3-minute intervals. To ensure full thermal stabilization, only the last two measurements out of the four were retained for analysis. Temperature at the end of the fourth measurement was recorded using a calibrated Physics 100-1 thermometer, with an uncertainty of ± 0.01ºC.

A total of 16 measurements performed using Sigma-Aldrich unpurified mCP dye, following the OVIDE-BOCATS protocol (CB'93+DVD'98; see Sect. 2.2.2), showed a consistent positive bias of **0.0105 ± 0.0013** $pH_T$ units relative to the nominal *TRIS* $pH_T$ values (Fig. 5). In contrast, 15 measurements conducted with the purified mCP dye, applying the L'11 parameterization, yielded values that were tightly centered around the nominal *TRIS* $pH_T$ values, with a negligible bias of **-0.0003 ± 0.0011** $pH_T$ units. No $\Delta R$ or $\Delta pH_T$ corrections were applied to either dataset, as the buffer capacity of *TRIS* is approximately 20 times higher than that of seawater, rendering the impact of the mCP dye addition on $pH_T$ negligible.

The correction of $_{434}A_{imp}$ due to mCP dye impurities, as proposed by DB'17, was evaluated using a value of $_{434}A_{imp}$ = 0.004413 absorbance units (given in their Table 2 for Sigma-Aldrich lot #11517KC at a mCP dye concentration of 3.3 µM in the seawater sample). Applying this correction enabled the use of the L'11 parameterization, substantially reducing the observed difference with purified mCP dye measurements to **0.0020 ± 0.0013** $pH_T$ units. Recognizing that impurities can vary between batches of the same mCP dye brand, a second test using $_{434}A_{imp}$ = 0.0034 absorbance units, consistent with that obtained by Álvarez et al. (2025), further reduced the offset to **-0.0005 ± 0.0013** $pH_T$ units.

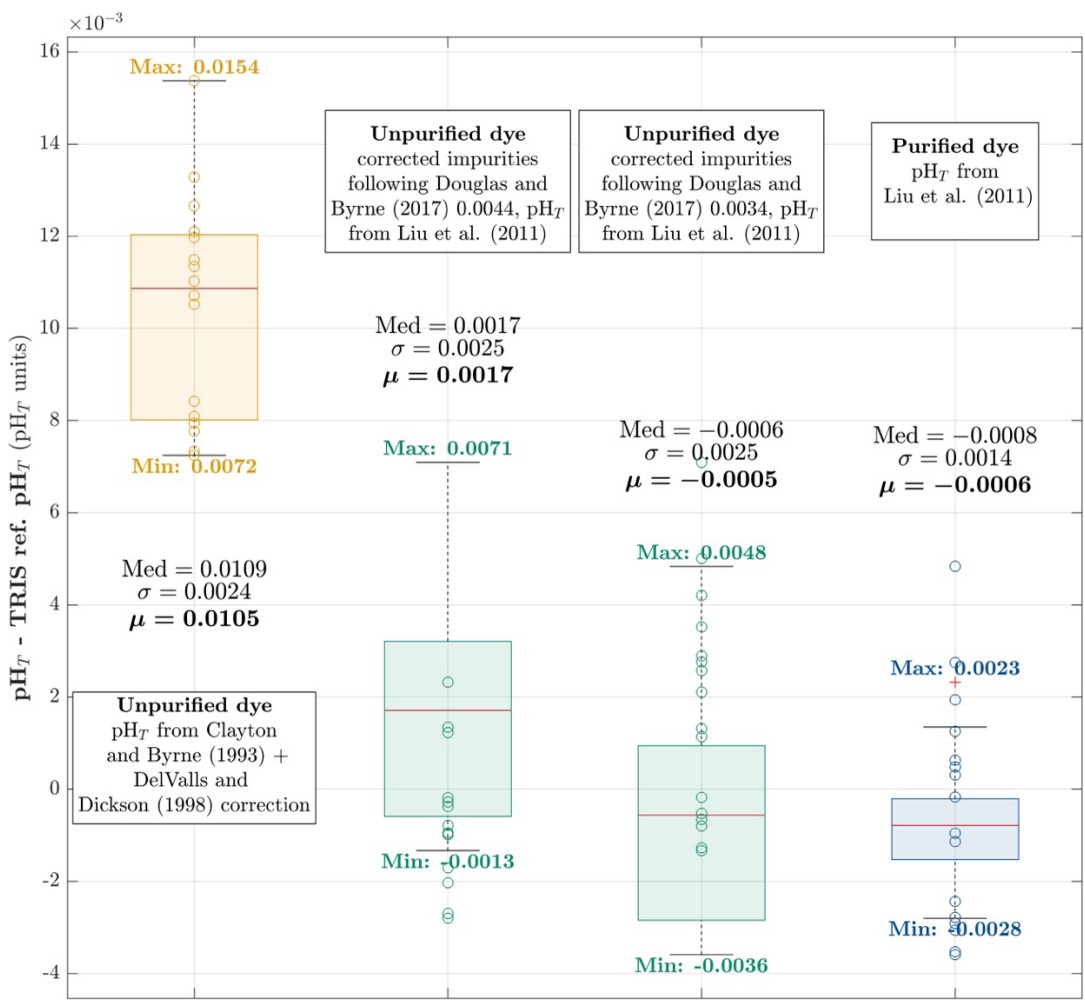

473

**Figure 5.** Whisker boxplots of $pH_T$ differences relative to *TRIS* buffer reference values, comparing results obtained using purified and unpurified mCP dyes. Each boxplot represents a different calculation approach: the orange boxplot shows $pH_T$ computed using unpurified mCP dye with the CB'93 parameterization and DVD'98 correction (N = 16); the blue boxplot shows $pH_T$ computed using purified mCP dye with the L'11 parameterization (N = 15); and green boxplots show $pH_T$ computed using unpurified mCP dye corrected for impurities using the DB'17 methodology with $_{434}A_{imp}$ = 0.0044 and $_{434}A_{imp}$ = 0.0034 absorbance units, respectively, and $pH_T$ then calculated using the L'11 parameterization. Within each boxplot, the red line indicates the median (*Med*), the box edges denote the first and third quartiles, and whiskers extend to data points within 1.5 times the interquartile range. Mean ($\mu$) and standard deviation ($\sigma$) values are provided for each subset, and the minimum and maximum values are indicated.

### 3.2 Duplicate measurements of natural seawater

During the 2021 cruise, six duplicate $pH_T$ samples were collected at each hydrographic station along the OVIDE-BOCATS transect, with sampling points carefully distributed throughout the water column to represent different water masses. In addition to the routine $pH_T$ determinations conducted according to the OVIDE-BOCATS protocol

(i.e., addition of 75 µL of Sigma-Aldrich [2 mM] mCP dye solution and $pH_T$ computed using CB'93+DVD'98; see
Sect. 2.2.2), these six replicate samples per profile were also measured using purified mCP dye (10 µL of [11 mM]
mCP dye supplied by Dr. Byrne's laboratory, and $pH_T$ determined using the L'11 parameterization).
These duplicate measurements encompassed the full $pH_T$ range typically observed during OVIDE-BOCATS
cruises (7.7 to 8.0 $pH_T$ units). As shown in Fig. 6, the standard OVIDE-BOCATS procedure yields $pH_T$ values that
are, on average, **0.0113 ± 0.0017** (N = 176) $pH_T$ units higher than those obtained using the purified mCP dye and
the L'11 parameterization. These results align with the findings from the *TRIS* buffer experiments (see Sect. 3.1).
The differences between the two methods showed no significant dependence on $pH_T$ (Fig. 7, green dots), in
agreement with the theoretical expectation illustrated in Fig. 2 (blue line, Aldrich-3.4).
When the DB'17 methodology was applied using a $_{434}A_{imp}$ value of 0.004413 absorbance units at $_{488}A = 0.225$, the
offset was reduced to 0.0015 ± 0.0017 $pH_T$ units—statistically indistinguishable from zero. Using a $_{434}A_{imp}$ value
of 0.0034 absorbance units further minimized the offset to **-0.0001 ± 0.0017** $pH_T$ units, corroborating the impurity
correction results from the TRIS buffer experiment. The individual $_{434}A_{imp}$ values applied here are proportionally
dependent on the $_{488}A$ in each measurement.
**3.3 Duplicate measurements of modified seawater**
During the OVIDE-2018 cruise, surface seawater with a salinity of 35.7 was treated using HCl or $Na_2CO_3$ to
produce four distinct $pH_T$ levels (7.45, 7.70, 7.95, and 8.19). These were stored in separate Niskin bottles and
sampled using the same protocol as for natural seawater. For each batch, four to six samples were analyzed for $pH_T$
using the standard OVIDE-BOCATS procedure (CB'93+DVD'98 and 75 µL of unpurified Sigma-Aldrich mCP
dye [2 mM] solution added to the cell; 5.36 µM final mCP dye concentration in the sample cell), and an equivalent
number was measured using purified mCP dye (75 µL of mCP dye [2.5 mM] solution added to the cell; 6.70 µM
final mCP dye concentration in the sample cell; FB5-2017 from Dr. Byrne's laboratory and the L'11
parametrization). This experiment extends the comparison conducted with *TRIS* buffer and natural seawater to a
wider $pH_T$ range, representative of conditions encountered in the South Atlantic and Pacific Oceans.
The mean $pH_T$ offset between the two methods (measuring with unpurified mCP dye and applying CB'93+DVD'98
versus purified mCP dye and applying the L'11 parameterization) across 22 measurements was **0.0109 ± 0.0011**
$pH_T$ units, consistent with previous results from *TRIS* and natural seawater duplicate samples (Sect. 3.1 and Sect.
3.2). No significant trend was observed across the $pH_T$ range (Fig. 7, regression p-level > 0.05). Similarly,
differences observed in natural seawater (green dots in Fig. 7) also showed no significant correlation with $pH_T$
(slope = -0.000 ± 0.002; p-level > 0.05). Furthermore, the magnitude and behavior of the observed differences are
consistent with theoretical expectations (Fig. 2, yellow line), assuming a $_{434}A_{imp}$ value of 0.0034 absorbance units
for Sigma-Aldrich mCP dye at a final concentration in the cell of 3.3 µM (as per Table 2 of DB'17).

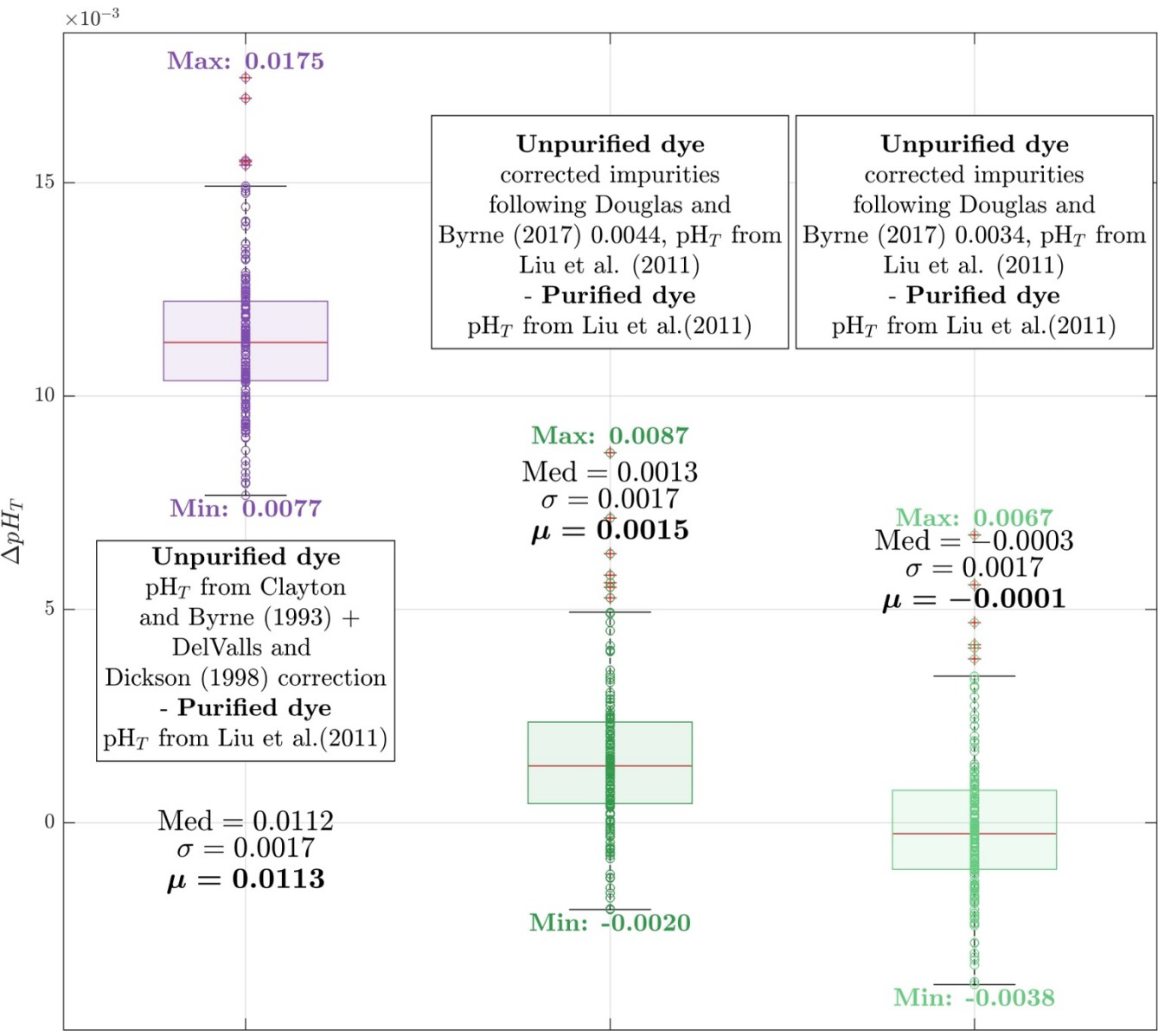

**Figure 6.** Whisker boxplots showing $pH_T$ differences between duplicate samples measured using purified and unpurified mCP dyes (N = 176). *Med* indicates the median (red line inside each box), $\sigma$ the standard deviation, and $\mu$ the mean of each subset. The minimum and maximum values are also indicated. The first boxplot (purple) shows the $pH_T$ difference between measurements using unpurified mCP dye ($pH_T$ calculated with CB'93+DVD'98) and purified mCP dye ($pH_T$ calculated with L'11 parametrization). The second boxplot (first green) shows differences after applying the DB'17 correction to R values obtained with unpurified mCP dye, using $_{434}A_{imp}$ = 0.0044 absorbance units, followed by $pH_T$ computation with L'11 parametrization. The third boxplot (second green) applies the same procedure but with $_{434}A_{imp}$ = 0.0034 absorbance units.

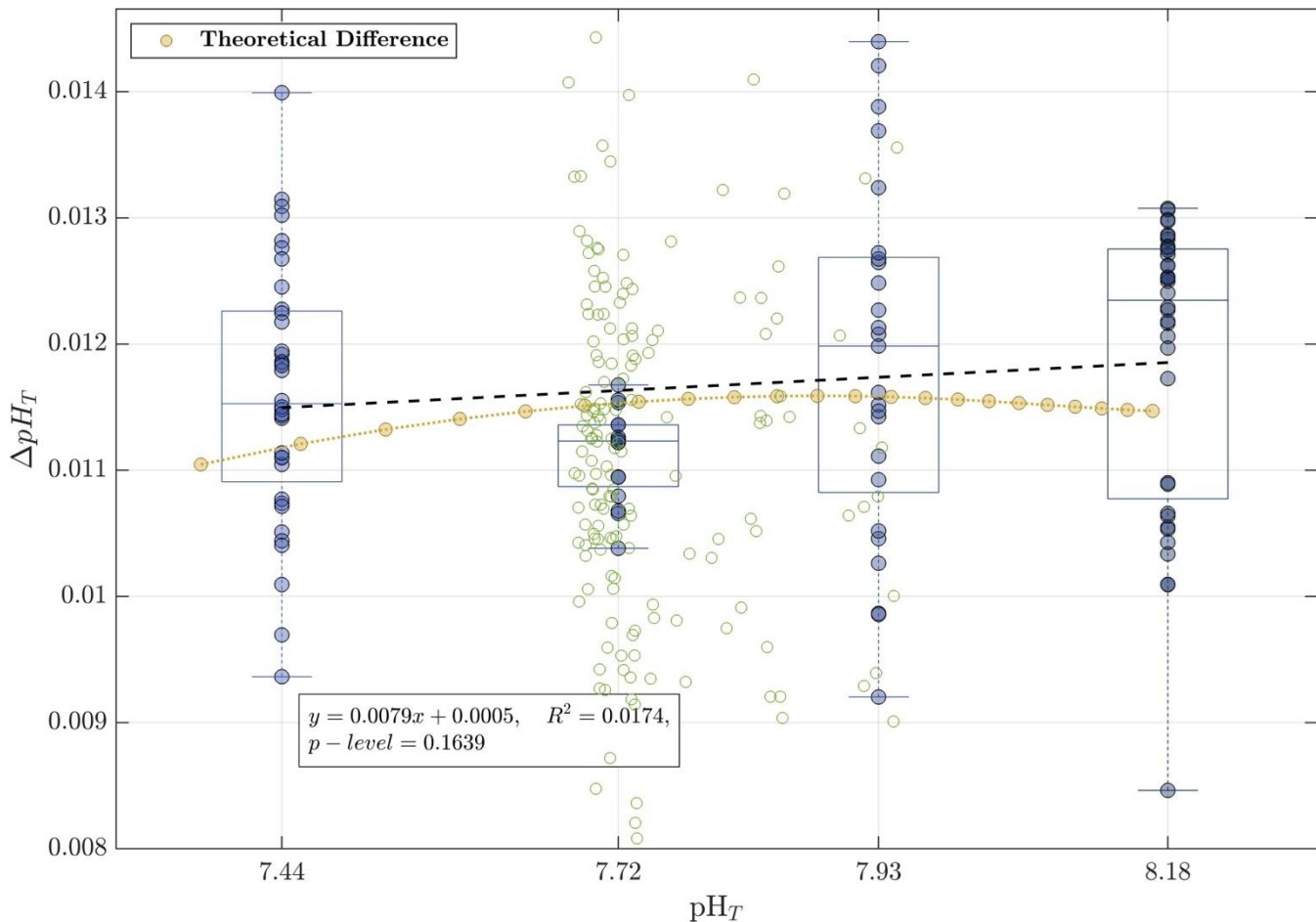

**Figure 7.** Differences in $pH_T$ measurements between replicate samples measured using unpurified mCP dye
(CB'93+DVD'98) and purified mCP dye (L'11 parameterization) across four $pH_T$ levels obtained from modified
seawater experiments. The black dashed line shows the linear regression of the differences as a function of $pH_T$,
with regression statistics summarized in the inset box. The yellow line represents the expected $pH_T$ differences
assuming a $_{434}A_{imp}$ value of 0.034 absorbance units, as modeled in Fig. 2. Green dots correspond to the 176
individual $pH_T$ differences presented in the first (purple) boxplot of Fig. 6.
**3.4 $pH_T$ correction**
DB'17 proposed a value of $_{434}A_{imp}$ = 0.004413 absorbance units for a Sigma-Aldrich mCP dye at a final
concentration of 3.3 µM in the sample cell ($_{488}A$ = 0.225; Takeshita et al., 2021). Based on our experimental results,
we found that a better fit under our conditions corresponded to a $_{434}A_{imp}$ value of 0.0034 absorbance units, which
is reasonable given that we are working with a different lot of mCP dye. Given that $_{434}A_{imp}$ is proportional to $_{488}A$,
which in turn reflects the final mCP dye concentration in the cell, we derived the following relationship:
$$_{434}A_{imp} = 0.0034 \,/\, 0.2250 \cdot {}_{488}A = 0.0151 \cdot {}_{488}A \qquad (8).$$
Accordingly, the impurity correction was applied to each sample by subtracting the computed $_{434}A_{imp}$ (Eq. 8) from
the sample measured $_{434}A$, yielding:

$$434A_{corr,pur} = 434A - 434A_{imp} = 434A - (0.0151 \cdot 488A) \qquad (9),$$

$$R_{corr,pur} = (578A - 730A) / (434A_{corr,pur} - 730A) \qquad (10),$$

where $434A_{corr,pur}$ is the corrected $434A$, and $R_{corr,pur}$ corresponds to the R value as if measured with a purified mCP dye. Subsequently, $pH_T$ was recalculated using the L'11 parameterization.

Implementing this correction required a comprehensive recovery and reassessment of historical $434A$, $578A$, and $488A$ values dating back to the 2002 cruise. Note that for data prior to 2018, $488A$ values were estimated from Eq. (7). While corrections related to the mCP dye addition effect were included in the data published in GLODAPv2.2023 (Lauvset et al., 2024), the $488A$-based correction described in Sect. 2.2.3 had not yet been incorporated. Final $pH_T$ values included in this work were computed from the corrected absorbance data, using the recalculated $R_{corr,pur}$, along with the updated mCP dye perturbation correction $\Delta(pH_T/488A)$-vs-$pH_{T,1}$.

## 4. Results and Discussion

### 4.1 Database product

We present a new database comprising 23,843 seawater samples with spectrophotometric $pH_T$ values, each accompanied by complete spatiotemporal metadata–including latitude, longitude, pressure, depth, date, and time– as well as in situ measurements of temperature, salinity, and dissolved oxygen. Dissolved oxygen concentrations were primarily determined using the Winkler titration method; where unavailable or deemed unreliable, values from a calibrated oxygen sensor mounted on the *Conductivity, Temperature, and Depth* (CTD) instrument were used. The dataset incorporates a quality flagging scheme consistent with GLODAPv2 recommendations (Key et al., 2015; Olsen et al., 2016), where flag 2 denotes good data (23,773 samples), 3 (19 samples) and 4 (51 samples) indicate questionable and bad data, respectively, and 9 denotes not measured.

Spectrophotometric $pH_T$ data collected between 2002 and 2018, available in the GLODAPv2.2023 release (Lauvset et al., 2024), were computed using the CB'93 parameterization with the DVD'98 correction (+0.0047 $pH_T$ units). The newly compiled $pH_T$ dataset presented here updates and corrects these data following the procedure presented in Sect. 3.4, applying the DB'17 adjustment and L'11 equation. In addition, the dataset significantly extends the temporal coverage by including $pH_T$ measurements from the 2021 and 2023 cruises, which were not previously available; these measurements are also corrected using the same procedure, resulting in a consistent final product across all cruises. In addition, for the first time, associated absorbance readings ($434A$, $578A$, and $488A$) are provided alongside $pH_T$ values. This comprehensive and corrected $pH_T$ dataset provides a robust foundation for future reassessments, such as the application of updated absorbance-to-$pH_T$ parameterizations or transformations to alternative pH scales (e.g., the "free" hydrogen ion scale).

### 4.2 Consistency of the $pH_T$ correction

The differences between the former-procedure—applying the CB'93 parameterization with the DVD'98 correction to R values derived from unpurified mCP dye—and the updated method presented here—applying the L'11 parameterization to R values derived from unpurified mCP dye corrected for the impurity effect (see Sect. 3.4)— are, on average, +0.011 ± 0.002 (1σ) $pH_T$ units (N = 23,843; Fig. S4 in the Supplement), in line with results from the assessment experiments described in Section *3*. These differences show a slight negative correlation with $pH_T$, with smaller offsets observed at higher $pH_T$ values (Fig. S4 in the Supplement). The slopes of the linear regressions of these differences versus $pH_T$ range from -0.0064 ± 0.0001 to 0.0005 ± 0.0001 $pH_T$ units. It should be noted that

this range is comparable in magnitude to other uncertainty sources—such as the effect of the addition of mCP dye
to the sample $pH_T$, and the instrumental measurement uncertainty (e.g., Sect. 2.2.3)—which may contribute to the
overall variability in the observed differences.
To assess the internal consistency and long-term comparability of the corrected $pH_T$ values, we examined the deep
layer of the Iberian Basin, associated with the North East Atlantic Deep Water (NEADW). This layer has been
recognized as a stable reference for the OVIDE-BOCATS program, as its properties show minimal long-term
variability (García-Ibáñez et al., 2016). Supporting this minimal variability, Steinfeldt et al. (2024) reported no
detectable accumulation of anthropogenic $CO_2$ in this layer based on chlorofluorocarbons measurements.
The average $pH_T$ in the NEADW layer over 11 cruises (2002–2023) was $7.7314 \pm 0.0015$ (1 standard deviation;
$1\sigma$), with only two cruises (2008 and 2018) exceeding the mean by more than 1 standard deviation (Fig. 8). The
standard deviation within individual cruises was generally low ($< 0.0018$ $pH_T$ units), with the exception of the 2004
cruise (0.0022 $pH_T$ units). Additionally, the cruise-specific mean $pH_T$ values showed no correlation with the
difference between the old and new $pH_T$ values (Fig. S5 in the Supplement). These findings reinforce the reliability
of the applied correction for the effects of impurities in the Sigma-Aldrich mCP dye (Sect. 3.4), based on a $_{434}A_{imp}$
value of $0.0034 \pm 0.0010$ absorbance units for $_{488}A = 0.225$.
However, it is important to acknowledge a limitation: the $_{434}A_{imp}$ value was not directly determined for each
individual batch of mCP dye, as recommended by DB'17. In practice, this is challenging—especially for older
cruises—since the specific mCP dye batches may no longer be available. While Sigma-Aldrich mCP dye batches
have been shown to have a narrow impurity range (typically $> 90\%$ purity; Álvarez et al., 2025), variations between
batches still exist. Thus, assuming a single correction value ($_{434}A_{imp} = 0.0034$) across all cruises could be questioned.
We conservatively estimate an absorbance uncertainty of $\pm0.001$ due to batch variability, which translates to an
uncertainty of approximately $\pm0.002$ $pH_T$ units. The estimate is consistent with the inter-cruise variability observed
(Fig. 8), supporting the use of $_{434}A_{imp} = 0.0034$ as a reasonable and robust correction value for the entire OVIDE-
BOCATS dataset.

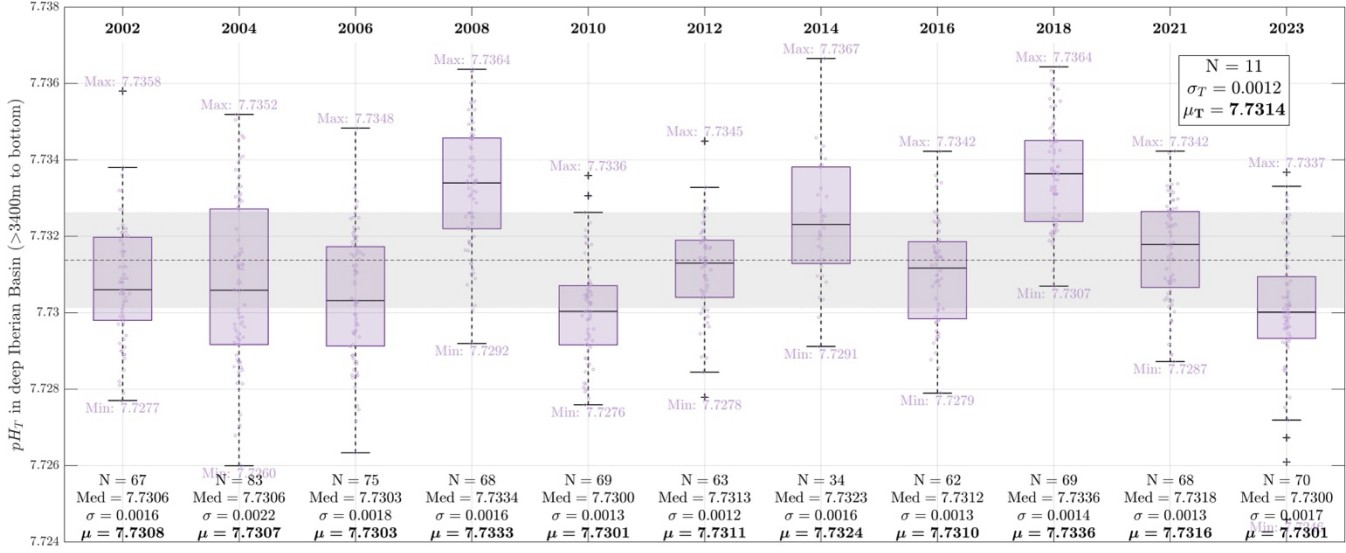

**Figure 8.** Variability of $pH_T$ in the lower NEADW (deep Iberian Basin, observations $> 3,400$ m and east of
15.45ºW) during the 11 OVIDE-BOCATS cruises. These data are used to assess the internal consistency of $pH_T$
measurements over time. $N$ refers to the number of samples analyzed in each cruise; $Med$ indicates the median $pH_T$
of each subset (shown as a black line within each boxplot); σ is the standard deviation; and μ is the mean. Minimum
and maximum values are also shown for each cruise. The overall mean ($\mu_T$; dashed horizontal line) and standard
deviation ($\sigma_T$; shaded gray band) across all 11 cruises are provided in the upper-right inset.

**4.3 Implications of the $pH_T$ correction on OA and aragonite saturation horizon**

The correction applied to our $pH_T$ dataset—an approximately constant offset across cruises—has negligible
implications for previously reported OA rates by García-Ibáñez et al. (2016) and Fontela et al. (2020a). However,
it does affect the calculation of the aragonite saturation, particularly the saturation horizon depth estimated by Pérez
et al. (2018) and García-Ibáñez et al. (2021). To assess this impact, aragonite saturation horizons were recalculated
using in situ temperature, salinity, $A_T$, and $pH_T$ values, both before and after correcting for $_{434}A_{imp}$ (see Sect. 3.4),
using the carbonate chemistry constants from Lueker et al. (2000) and the boron formulation of Lee et al. (2010).
This reevaluation reveals a more pronounced reduction in aragonite saturation at the surface (from -0.040 to -
0.065), relative to pre-industrial conditions, which progressively diminishes with depth, reaching changes of -0.016
near the seafloor. Although changes at depth appear small in absolute terms, the weak vertical gradient in aragonite
saturation in deeper layers translates into a significant vertical shift in the saturation horizon—rising by
approximately 120 m to 200 m. Based on the reproducibility of the doubloons (Sect. 2.2.5) and the standard
deviation of the mean $pH_T$ in the NEADW layer over 11 cruises (Sect. 4.2), the uncertainty in the saturation-depth
change is estimated at 17 meters, while the uncertainty in the aragonite saturation state is 0.0033 units. For instance,
in the NA subpolar gyre, where the aragonite saturation horizon currently resides near 2,700 m depth, the revised
(lower) $pH_T$ values shift it upward by approximately 150 m. This shift implies that vulnerable cold-water coral
ecosystems may be exposed to undersaturation conditions in shallower and more extensive regions than previously
estimated. This reassessment underscores the importance of accurate $pH_T$ determinations: even subtle biases can
propagate into substantial differences in projected impacts on sensitive deep-sea habitats.
To investigate the spatial and temporal evolution of $pH_T$ along the OVIDE-BOCATS section, observations from
the new OVIDE-BOCATS database were interpolated onto a common 7 km x 1 dbar grid. Each cruise's station
positions were projected onto the grid by identifying the closest grid node (minimum distance), followed by linear
interpolations using a Delaunay Triangulation approach (Amidror, 2002). This method ensured optimal station
overlap while preserving dataset consistency across years.
Figure 9 displays the $pH_T$ distributions from the 11 cruises (2002–2023), along with the overall mean distribution.
Surface waters show the highest $pH_T$ values, particularly along the eastern boundary, where elevated temperatures
(not shown) partly contribute to the increase. Minimum $pH_T$ values generally occur in intermediate waters (~500–
1,500 m), except in the Iberian Basin, where the presence of Mediterranean Water–characterized by a warm, saline
core at ~1,000 m–causes a downward shift of the $pH_T$ minimum to ~2,000 m.
Notably, a persistent $pH_T$ minimum appears in the Iceland Basin between 500 m and 1,000 m, associated with
intermediate waters with high Apparent Oxygen Utilization (AOU; Fig. S6a in the Supplement; Lauvset et al.,
2020). This layer, influenced by older water masses transported by the North Atlantic Current (NAC) and exhibiting
elevated remineralization rates (de la Paz et al., 2017), has shown significant spatial expansion over time. Since
2016, waters with $pH_T$ below 7.71 have progressively expanded eastward, deepening toward ~2,000 m and reaching
the Azores-Biscay Ridge (see Fig. 1 for georeference). In addition, the low-$pH_T$ layer has spread into the Irminger
Basin since 2010.

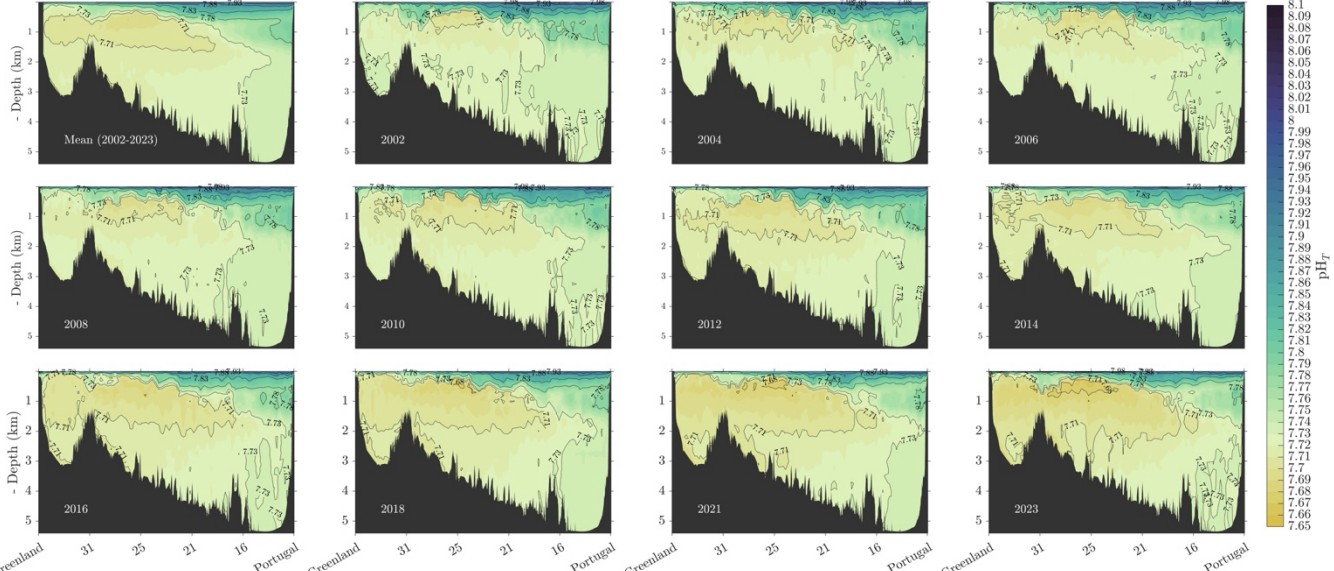

**Figure 9.** Distribution of $pH_T$ normalized to 25ºC and 1 atm along the OVIDE-BOCATS section (Fig. 1) for each cruise from 2002 to 2023, as well as the overall mean of all 11 OVIDE-BOCATS cruises**.** The section is plotted with longitudes (ºW) in the x-axis.

In the Irminger Basin, a subsurface $pH_T$ minimum (< 7.73) was already evident in 2002, associated with DSOW. This signal reappeared in 2014 with even lower values (< 7.70) and has progressively thickened through to the most recent observations in 2023. Although less intense than the DSOW signal, ISOW in the Iceland Basin has also shown a noticeable OA signal since 2016, reaching comparable $pH_T$ values to those found in DSOW. In contrast, the deep $pH_T$ maximum observed in 2002 at depths of 2,700—3,000 m—extending eastward from 20ºW— had largely disappeared by 2010, becoming confined to the Iberian Basin. There, maximum $pH_T$ values persist between 3,000 and 5,250 m depth, corresponding to the core of NEADW.

This contrast in deep-ocean $pH_T$ between the more recently ventilated waters of the Irminger and Iceland Basins and the older, more stable NEADW is consistent with differential exposure to $C_{ant}$. Waters in the subpolar basins— having had more recent contact with the atmosphere or mixed with recently ventilated layers—have absorbed more $C_{ant}$, leading to their enhanced OA (Fig. 10) (García-Ibáñez et al., 2016).

DSOW exhibits a particularly strong OA (rate < -0.0015 $pH_T$ yr$^{-1}$), extending along the bottom of the Irminger Basin. A similar, though less intense, signal is seen in ISOW (< -0.0010 $pH_T$ yr$^{-1}$). The highest OA rates (< -0.002 $pH_T$ yr$^{-1}$) are observed in the surface layers (0—500 m) due to direct air-sea $CO_2$ exchange, consistent with the rate reported in Reverdin et al. (2018) for surface data in the NA subpolar gyre. These upper layers also exhibit high interannual $pH_T$ variability (Fig. S7 in the Supplement), which correlates negatively with AOU (Fig. S6b in the Supplement). This $pH_T$-AOU relationship suggests a strong influence of mesoscale variability, particularly associated with the NAC, which is known for its energetic and variable meandering in this region (Daniault et al., 2016). As a result, OA rates in NAC-influenced areas may be either enhanced or masked by spatial variability of the natural component in the $pH_T$ variability, sometimes leading to non-significant OA rates despite ongoing $C_{ant}$ uptake.

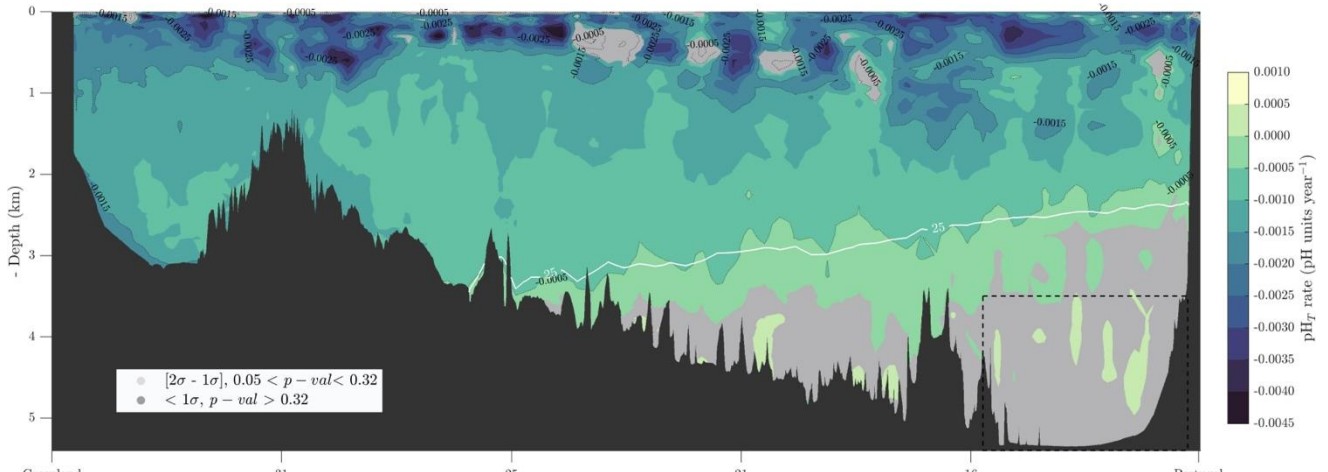

**Figure 10**. Linear trends in $pH_T$ at 25ºC and 1 atm from 2002 to 2023 (i.e., OA rates) along the OVIDE-BOCATS
section (Fig. 1), based on high-resolution interpolations. Blue (yellow) shading indicates more (less) negative OA
rates. Grey areas denote trends that are not statistically significant at the 1σ level. The white contour represents the
silicate isoline of 25 µmol kg$^{-1}$, and the dashed black box marks the deep Iberian Basin region used for measurement
quality-control (see Sect. 4.2). Longitude (º W) is shown on the x-axis.
On the other hand, in the deep-water masses east to 30ºW, there is a clear transition to non-significant OA rates at
depth (Fig. 10). This transition coincides with the 25 µmol kg$^{-1}$ silicate isoline, which marks the boundary between
NA-origin waters and those with Antarctic influence (García-Ibáñez et al., 2015). At this silicate level, García-
Ibáñez et al. (2015) estimated that NEADW accounts for ~30% of the water mass composition, while LSW and
ISOW contribute the remaining 70%, with proportions varying with depth. NEADW originates from Antarctic
Bottom Water, which flows into the Eastern North Atlantic Basin from the Vema Fracture Zone (Mercier and
Morin, 1997), and is largely devoid of $C_{ant}$ (Steinfeldt et al., 2024). Consequently, as silicate concentrations increase
beyond 25 µmol kg$^{-1}$, the influence of NEADW becomes dominant, resulting in non-significant OA rates (Fig. 10),
and elevated AOU values (Fig. S6a in the Supplement), consistent with the advanced age and limited ventilation
of these waters.
**5. Conclusions**
We present a new, rigorously quality-controlled dataset of discrete spectrophotometric $pH_T$ measurements from the
North Atlantic, spanning over two decades and including absorbance data. This dataset provides a unique resource
for the ocean carbon research community, enabling retrospective reassessment of $pH_T$ values and derived variables
under updated methodological standards.
Our analysis revealed that $pH_T$ values measured with an unpurified mCP dye from Sigma-Aldrich exhibit a
consistent positive bias of +**0.011 ± 0.002** $pH_T$ units, on average, compared to those measured using purified mCP
dye, with this offset decreasing slightly at higher $pH_T$. While the correction applied has negligible influence on
previously published OA trends, it significantly affects derived variables such as the aragonite saturation horizon,
which is now estimated to be up to 200 m shallower in certain regions. These changes have implications for
assessing the vulnerability of deep-sea ecosystems to OA and underscore the need for highly accurate $pH_T$
measurements.

Our results reinforce findings from recent studies (e.g., Carter et al., 2024a; Takeshita et al., 2021, 2022) and support the following recommendations:

1. Ideally, $pH_T$ measurements should be carried out using well-characterized, purified mCP dyes and following consensus procedures that ensure SI traceability (Capitaine et al., 2023; Carter et al., 2024a), regardless of mCP dye used.

2. Although the correction applied here ($_{434}A_{imp}$ = 0.0034) yielded consistent results, we recommend the determination of batch-specific $_{434}A_{imp}$ values (Douglas and Byrne, 2017; Álvarez et al., 2025).

3. The effect of mCP dye addition on sample $pH_T$ is comparable in magnitude to spectrophotometer non-linearity. Our findings support the estimation of this effect via $\Delta(pH_{T/488A})$-vs-$pH_{T,1}$ approach proposed by Takeshita et al. (2022), and are consistent with recommendations by Li et al. (2020).

4. While the mCP dye does not significantly alter the *TRIS* buffer $pH_T$, accurate temperature control is essential. *TRIS* remains suitable for methodological validation, with spectrophotometer behavior being the primary concern (Capitaine et al., 2023).

## 6. Data availability

The complete OVIDE-BOCATS $pH_T$ dataset presented in this study is made available at https://doi.org/10.5281/zenodo.17789895 (Pérez et al., 2025) in multiple formats to ensure broad accessibility and compatibility with different research workflows. The dataset includes 23,843 spectrophotometric $pH_T$ measurements along with associated absorbance data ($_{434}A$, $_{578}A$, and $_{488}A$) and complete spatiotemporal metadata from 11 cruises spanning 2002-2023. Data are provided as: (1) comma-separated values (CSV) format for general use, (2) WHP-Exchange bottle format following WOCE Hydrographic Program Exchange format standards, (3) NetCDF format with CF-compliant metadata, and (4) Apache Parquet format with both CF standard names and the proposed metadata conventions of Jiang et al. (2022). This multi-format approach ensures the data can be readily integrated into existing oceanographic databases and analysis workflows, adhering to FAIR (Findable, Accessible, Interoperable, and Reusable) data principles. All formats include quality flags for $pH_T$ following GLODAP recommendations.

## Supplementary Information

The supplement related to this article is available online at:

## Author contribution

FFP, MLM, and AV designed the study, conceptualization, methodology, validation and formal analysis. AV, PL and FFP give supervision, administration and funding. The manuscript was written by MLM and FFP and edited, and also revised by MGI and discussed by all authors. The dataset, data curation and validation were done by FFP, AV, MA and MLM.

## Competing interests

Author A. Velo is a member of the editorial board of the journal.

765

## Acknowledgments

We thank all of the scientists, technicians, personnel, and crew who were responsible for the collection and analysis of the over 23 000 samples included in the final dataset.

## Financial support

F.F. Pérez and X.A. Padín were supported by FICARAM+ (PID2023-148924OB-I00) project funded by MICIU/AEI/10.13039/501100011033 and FEDER, UE. A. Velo was supported by the European Union under grant agreement no. 101094690 (EuroGO-SHIP). M. López-Mozos was supported by the grant PRE2020-093138 funded by MICIU/AEI/10.13039/501100011033 and by "ESF Investing in your future". M. Fontela was funded by PTA2022-021307-I, MCIN/AEI/10.13039/501100011033 and by FSE+. M.I. García-Ibáñez and M. Álvarez were supported by the ESMARES (Marine Strategy Directive Framework) program funded by the Spanish Ministry for the Ecological Transition and the Demographic Challenge. N.M. Fajar was supported the Complementary Science Plans for Marine Science of 'Ministerio de Ciencia, Innovación y Universidades' included in the Recovery, Transformation and Resilience Plan (PRTR-C17.I1) funded through 'Xunta de Galicia' with Next Generation EU and the European Maritime Fisheries and Aquaculture Fund.

This work has been supported by FICARAM+ (PID2023-148924OB-I00) project funded by MICIU/AEI /10.13039/501100011033 and FEDER, UE.

This work was co-funded by the European Union under grant agreement no. 101094690 (EuroGO-SHIP) and UK Research and Innovation (UKRI) under the UK government's Horizon Europe funding guarantee [grant numbers: 10051458, 10068242, 10068528]. Views and opinions expressed are however those of the author(s) only and do not necessarily reflect those of the European Union or European Research Executive Agency. Neither the European Union nor the granting authority can be held responsible for them.

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
