# Peer review of "Two decades of pHT measurements along the GO-SHIP A25 section in the North Atlantic"

_Earth System Science Data, 2025_

## Author Comment (AC1)

**Anonymous Referee #1,** 26 Oct 2025
 General comment:

The North Atlantic Ocean is an important CO2 sink (Takahashi et al. 2009) and this region contains high concentrations of anthropogenic CO2 in the water column (Khatiwala et al., 2013; Steinfeldt et al, 2024). Thanks to numerous observations in this basin since the seventies, it is now well known that simulated carbonates systems properties, including CO2 fluxes, OA and Cant, present significant bias. Quoting Perez et al., 2024: "The largest disagreement in the CO2 flux between GOBMs and pCO2 products is found north of 50°N". Bias between models and data-based estimates are also observed for the Cant inventories in the North Atlantic (Perez et al, 2024, their figure 7). This calls for new analysis based on series of cruises to investigate the Cant variability, from seasonal to multi-decadal, such as recently presented by Bajon et al (2025). In this context, it is also important to extend and/or revise the data when observational bias are suggested or identified (e.g. Wang et al , 2025 for oxygen). Here the authors suggest that a correction of their pH data obtained along the OVIDE-BOCATS sections should be applied (by 0.011 on average). As an example, the comparison of data presented in the NEADW is convincing. Although this has apparently no impact on the pH trends, they show that the correction leads to a shift for the ASD. This new dataset (including 2021 and 2023 cruises) represents an important product, not only to re-investigate the drivers of ASD (Lauvset et al, 2020) but also for comparisons of pH data from BGC-Argo floats in this region (Wimart-Rousseau et al, 2024) as well as for model validation. The dataset includes 23500 corrected pH data from 11 cruises that will be probably revisited in the next GLODAP version. I wondered why authors did not include their AT data in the file that would help to calculate CT and Cant as well.

The document is well structured, figures and tables adapted. I recommend publication after minor revision. Below are listed specific comments.

*We would like to thank Anonymous Referee #1 for the thoughtful review and the constructive feedback provided. We have carefully considered all comments, as detailed below. We specifically address the question regarding the AT data in our response below.* Anonymous Referee #1's evaluation is reproduced in black, *the author's responses appear in green and italics, the original manuscript text appears in black and italics,  and* the changes introduced in the manuscript are shown in blue and italics.

Specific comments:

**C-01**: Title: "Two decades of pHT measurements along the GO-SHIP A25 section". For readers not familiar with GO-SHIP and cruises numbers, maybe specify this is in the North Atlantic: Two decades of pHT measurements along the GO-SHIP A25 section in the North Atlantic.  *We agree, corrected.*

**C-02**: Line 59: Not sure that Ishii et al (2025) is a correct reference for OA and BGC-Argo data. *Deleted, thank you.*

**C-03**: Line 139: "as well as the ocean's capacity to absorb, store, and transport CO2 (Pérez et al., 2013; Zunino et al., 2015)." You can add reference to Bajon et al, (2025) when published.

*Thank you for the reference. We have included as follows: "(...) as well as the ocean's capacity to absorb, store, and transport $CO_2$ (Bajon et al., 2025, manuscript in review; Pérez et al., 2013; Zunino et al., 2015)."*

**C-04: Line 141:** "and understanding the SPNA's response to climate change (Rodgers et al., 2023)." Is it the correct reference for the response to climate change ? DeVries et al, (2023) would be more appropriate.

*The article by Rodgers et al. (2023) discusses and focuses on the challenges that GOBMs face in modeling the impact of the biological pump on $CO_2$ fluxes in subpolar regions, so we prefer to keep this citation. However, we agree that DeVries et al. (2023) is also highly relevant, and we have included it as well as: "(DeVries et al., 2023; Rodgers et al., 2023)."*

**C-05:** Figure 1: I guess one of the cruise in 2014 (GEOVIDE) extended to the west off Greenland (stations south of Labrador Sea). Are these stations included in the new dataset ? If yes, this should be shown in figure 1.

*Absolutely, these stations are included in the dataset. However, the previous version of Figure 1 displayed only the stations along the A25 line, and this was not indicated in the caption. We apologize for the oversight. We have revised Figure 1 to include all stations included in the dataset, and we have updated the caption accordingly: "Figure 1. Bathymetric map showing the main water masses and circulation patterns within the SPNA region covered by the OVIDE-BOCATS program (station locations indicated by black dots). Stations outside the main OVIDE-BOCATS (A25) section are opportunistic stations, dependent on the ship's route, whose data are also included in the final database product: "plus" symbols near Iceland and Greenland are stations sampled in 2006; "pentagon" and "rhombus" located in the Labrador Sea are stations sampled during 2012 and 2014 cruises; and "asterisk" symbols are stations sampled in 2023, near Greenland and in the Irminger Sea. The inset (…)".*

**C-06: Line 176:** "…should be adjusted by +0.0047 pHT units (Lee et al., 2000)." I think the reference should be DelValls and Dickson (1998). Lee et al used this correction to revise the dissociation constants.

*Thank you for pointing this out. Lee et al. (2000) explicitly stated that "consequently, spectroscopic pH values obtained in the laboratory [Lee el al., 1996] and in field studies [Millero el al., 1993; Claylon el al., 1995; Lee el al., 1997] may need to be revised upward by 0.0047 pH units.". However, we agree with Anonymous Referee #1 that DelValls and Dickson (1998) is the most appropriate reference to cite for this correction. Therefore, we have retained the reference to Lee et al. (2000) and added DelValls and Dickson (1998). The revised sentence now reads: "These corrected TRIS $pH_T$ values have recently been confirmed by Müller et al. (2018). Consequently, spectrophotometric*

*pH_T values obtained using the CB'93 parameterization should be adjusted by +0.0047 pH_T units (DVD'98; Lee et al., 2000)."*

**C-07: Line 188:** "(see Fig. S1 in Álvarez et al.,submitted)." The paper by Álvarez et al. is not available at that stage (but would be happy to read it).

*We regret that it is not yet accessible. The paper is currently accepted, and we expect it to become available soon. Here is the Figure S1 kindly provided by M. Álvarez.*

[Figure]

***Supporting Information Figure S1.*** *Graphical representation of the terms in Eq. 1 according to the four different parameterizations CB93 (Clayton and Byrne 1993), LIU11 (Liu et al, 2011), LOU17 (Louicaides et al., 2017), MR18 (Muller and Rehder 2018) (See Supporting Information Table S1). Parameterizations and bibliographic references are given in Supporting Information Table S1. The terms (A) $pK^T_2e_2$, (B) $e_3/e_2$ (where the parameterization by DG14 is also represented), and (C) $e_1$ are represented as a function of salinity and referred to 25ºC. Plot (D) shows the composite term $log_{10}((R-e_1)/(1-R\cdot e_3/e_2))$, which is added to (A) $pK^T_2e_2$ to obtain (E) seawater pH referred to 25ºC as a function of the absorbance ratio (R) and salinity. Plot (F) shows the pH difference between parameterizations CB93, LOU17, and MR18 and that of LIU11 at 25ºC, and salinity of 35. All pH values are provided on the total hydrogen ion scale.*

**C-08: Line 387**: "This fit, based on 6,910 samples from the 2018, 2021, and 2023 cruises (Supporting Information Fig. 3)". In Figure S3, N= 2673. Is the fit based on 6910 or 2673 samples ?

*We apologize for the confusion. There was a typo in the legend of Supporting Information Figure 3. The fit is indeed based on 6,910 samples. We have corrected the figure legend accordingly. Thank you for catching this oversight.*

**C-09: Line 530:** "While corrections related to the mCP dye addition effect were included in the data published in GLODAPv2.2023 (Lauvset et al., 2024), the 488A-based correction described in Sect. 2.2.3 had not yet been incorporated". This is an important information for those who used and will use GLODAPv2.2023. Maybe also indicate here if the most recent cruises (2021 and 2023) have been submitted to GLODAP for the next version ?

*The 2021 and 2023 cruise data have not yet been submitted to GLODAPv2. The submission of the data has been delayed while we prepared this study, in order to avoid duplicating the submission of incomplete datasets. The dataset will be finalized with the measured and interpolated total alkalinity values and deposited in the SEANOE and DIGITAL-CSIC repositories under a single DOI. They will then be available for integration into GLODAP*

**C-10: Line 536**: "We present a new database comprising 23,535 seawater samples with spectrophotometric pHT values,…". I wondered why authors did not include their AT data. **C-11: Line 597**: "To assess this impact, aragonite saturation horizons were recalculated using in situ temperature, salinity, AT, and pHT values,…". Interesting sensitivity test, but the AT data are not in the files (correct ?).

*Correct, the total alkalinity (AT) data is not included in the current database files. The AT sampling strategy of the OVIDE-BOCATS cruises differed of pH: AT samples were collected only every two stations and with fewer samples per profile, which means that different processing is required before the two datasets can be combined. We plan to develop an interpolating approach to estimate AT at the pH sampling resolution while preserving the natural variability of the AT field. This work will evaluate different interpolation methods and proxies (e.g., salinity-based relationships, neural-network approaches). On the other hand, AT data from 2002 to 2018 are already available through GLODAP, where they have undergone extensive quality control. The AT data from the 2021 and 2023 cruises will soon be accessible via SEANOE and DIGITAL.CSIC.*

**C-12: Line 600:** "This reevaluation reveals a more pronounced reduction in aragonite saturation at the surface (from -0.040 to -0.065),…". I suspect this is reduction from pre-industrial period. Please clarify.

*Yes, this reduction refers to the change relative to pre-industrial conditions. We have clarified it in the text as follows: "This reevaluation reveals a more pronounced reduction in aragonite saturation at the surface (from -0.040 to -0.065), relative to pre-industial conditions, (...)".*

**C-13: Line 622:** "Notably, a persistent pHT minimum appears in the Iceland Basin between 500 m and 1,000 m, associated with intermediate waters with high Apparent Oxygen Utilization". On this topic, the impact of biological processes on pH distribution was quantified by Lauvset et al (2020).

*Thank you, reference included.*

**C-14: Line 651:** "The highest OA rates (< -0.002 pHT yr-1) are observed in the surface layers (0—500 m) due to direct air-sea CO2 exchange." Interestingly, such rate was also deduced in the NASPG from other surface data (-0.0021 pHT yr-1, Reverdin et al, 2018).

*Thank you for this remark. We have added this contextual information as follows: "The highest OA rates (< -0.002 $pH_T$ $yr^{-1}$) are observed in the surface layers (0—500 m) due to direct air-sea $CO_2$ exchange,* consistent with the rate reported by Reverdin et al. (2018) for surface waters in the NA subpolar gyre."

**C-15: Line 652**: "These upper layers also exhibit high interannual pHT variability (Fig. 10), which correlates negatively with AOU (Supporting Information Fig. 6b)." Figure 10 does not show the interannual variability. Maybe refer to Figure 9 here or present another figure for surface layer (in Supp Mat ?).

*Thank you for this suggestion, we agree. We have included a new figure to the Supporting Information (Supporting Information Fig. 7), which shows the standard deviation of $pH_T$ across the 11 cruises (panel A) and the standard error of the $pH_T$ rate (panel B). We have updated the text to direct the reader to this new figure as follows: "These upper layers also exhibit high interannual $pH_T$ variability ( Supporting Information Fig. 7), which correlates negatively with AOU (Supporting Information Fig. 6b)."*

**C-16: Line 673:** "NEADW originates from Antarctic Bottom Water, formed in the Vema Fracture Zone, and is largely devoid of Cant (Steinfeldt et al., 2024)." See also Mercier and Morin (1997) who first investigate the AABW in the Atlantic Fracture Zones.

*Thank you for this valuable reference. We have included it into the revised text as follows: "NEADW originates from Antarctic Bottom Water,  which flows into the Eastern North Atlantic Basin from  the Vema Fracture Zone (Mercier and Morin, 1997), and is largely devoid of $C_{ant}$ (Steinfeldt et al., 2024). "*

**C-17:** "6. Data availability" I wanted to explore the files but unfortunately, no access. On "zenodo" the message is: The record is publicly accessible, but files are restricted to users with access.

*We sincerely regret that you were unable to access the dataset. When submitting the article, following the options offered by ESSD, we opted to keep the data in the permanent repository (DOI Zenodo) under embargo until the article's publication. Simultaneously, we provided a temporary public repository containing the exact same content for reviewers. This temporary link is: https://saco.csic.es/s/kKqDXFYGKsKbaXj. According to ESSD procedures, this link should have been sent upon acceptance of the referees. We apologize for the oversight and any inconvenience it caused.*

Reference added in this review, not listed in the MS:

Bajon, R., Carracedo, L. I., Mercier, H., Asselot, R., and Pérez, F. F.: Seasonal to long-term variability of natural and anthropogenic carbon concentrations and transports in the subpolar North Atlantic Ocean, EGUsphere [preprint], https://doi.org/10.5194/egusphere-2025-4425, 2025.

DeVries, T., Yamamoto, K., Wanninkhof, R., Gruber, N., Hauck, J., Müller, J. D., et al. (2023). Magnitude, trends, and variability of the global ocean carbon sink from 1985 to 2018. Global Biogeochemical Cycles, 37, e2023GB007780. https://doi.org/10.1029/2023GB007780

Khatiwala, S., Tanhua, T., Mikaloff Fletcher, S., Gerber, M., Doney, S. C., Graven, H. D., et al. (2013). Global ocean storage of anthropogenic carbon. Biogeosciences, 10(4), 2169–2191. https://doi.org/10.5194/bg-10-2169-2013

Lauvset, S. K., Carter, B. R., Perez, F. F., Jiang, L.-Q., Feely, R. A., Velo, A., & Olsen, A. (2020). Processes driving global interior ocean pH distribution. Global Biogeochemical Cycles, 34, e2019GB006229. https://doi.org/ 10.1029/2019GB006229

Mercier, H and Morin, P: Hydrography of the Romanche and Chain Fracture Zones, 1997 JOURNAL OF GEOPHYSICAL RESEARCH-OCEANS, VL 102, 10373, DI 10.1029/97JC00229

Reverdin, G., Metzl, N., Olafsdottir, S., Racapé, V., Takahashi, T., Benetti, M., Valdimarsson, H., Benoit-Cattin, A., Danielsen, M., Fin, J., Naamar, A., Pierrot, D., Sullivan, K., Bringas, F., and Goni, G.: SURATLANT: a 1993–2017 surface sampling in the central part of the North Atlantic subpolar gyre, Earth Syst. Sci. Data, 10, 1901-1924, https://doi.org/10.5194/essd-10-1901-2018, 2018.

Takahashi, T., et al, 2009. Climatological Mean and Decadal Change in Surface Ocean pCO2, and Net Sea-air CO2 Flux over the Global Oceans. Deep-Sea Res II, doi:10.1016/j.dsr2.2008.12.009

Wang, Z., et al, 2025. Bias Evaluation for Sensor-Based Dissolved Oxygen from CTD and Profiling Floats in the World Ocean Database JOURNAL OF ATMOSPHERIC AND OCEANIC TECHNOLOGY, 42, DOI: 10.1175/JTECH-D-25-0027.1

Wimart-Rousseau, C., Steinhoff, T., Klein, B., Bittig, H., and Körtzinger, A.: Technical note: Assessment of float pH data quality control methods – a case study in the subpolar northwest Atlantic Ocean, Biogeosciences, 21, 1191–1211, https://doi.org/10.5194/bg-21-1191-2024, 2024.

---

## Author Comment (AC2)

2025-11-14

**Anonymous Referee #2**, 14 Nov 2025

General Comments

I really enjoyed reading this paper. It is well written, thoughtfully laid out, and has incorporated many rigorous experiments explained in significant detail. I liked the background information provided for the A25 section and felt it gave good context to the remainder of the manuscript. This large pHT dataset will be a great resource for researchers. I particularly commend the authors for including the absorbance values in their pHT dataset, which the authors correctly point out will be useful in the future if pHT characterizations are updated. I also appreciate that the authors put these pHT revisions in the context of the larger scale OA trends. I recommend acceptance of the manuscript after some minor revisions.

*We would like to thank Anonymous Referee #2 for the thoughtful review and the constructive feedback provided. We are pleased that the referee found our work valuable and engaging. All comments have been carefully considered, as detailed in the responses below.*

Anonymous Reviewer #2's evaluation is reproduced in black, *the author's responses appear in green and italics,* the original manuscript text appears in black and italics, *and the changes introduced in the manuscript are indicated in blue and italics.*

Specific Comments

**Lines 74-75:** pHT calculated from AT and CT is known to have potential issues, such as the pH-dependent pH offset and larger uncertainty. I would argue that this sentence should be rephrased to convey that both (1) pHT reported with no additional internal consistency corrections AND (2) pHT calculated from AT and CT could both affect the reliability of OA analyses. The key is having directly measured pHT that can be "trusted".

*Thank you for the appropriate suggestion, we agree with your comment. We have rephrased the text as follows: "$A_T$ and $C_T$ measurements are generally considered more reliable due to the availability of standardized reference materials, consensually accepted methods, and quality control procedures. In contrast, although $pH_T$ measurements are technically precise, easy to perform, and cost-effective, their intercomparability is more challenging, arising from methodological inconsistencies across various research initiatives (Dickson et al., 2015; Ma et al., 2019; Álvarez et al., 2020; Capitaine et al., 2023). The lack of traceability to a common reference, preferentially the International System of units for spectrophotometric pH measurements (Dickson et al., 2015), and the unavailability of pH reference materials within the seawater pH range (Capitaine et al., 2023), together with the documented issues affecting $pH_T$ calculated from $A_T$ and $C_T$ (including $pH_T$-dependent offsets and larger propagated uncertainties; Álvarez et al., 2020; Carter et al., 2024b), mean that neither unadjusted direct observations nor calculated values currently provide a fully trusted global reference. Both limitations may therefore affect the reliability of pH data for climate-quality OA assessments.*

*as originally reported with no additional internal consistency corrections, a limitation that may potentially affect the reliability of OA analysis from directly measured pH$_T$ data instead of those calculated from A$_T$ and C$_T$ (Álvarez et al., 2020; Carter et al., 2024b).''*

**Lines 78-79:** The pH method was detailed previous to the 1990s (Robert-Baldo et al. 1985; Byrne and Breland, 1989), using mCP as the indicator dye began in the 1990s.

*Thank you. We have rephrased the text as follows: "Briefly, the spectrophotometric pH method is a straightforward technique that involves adding an acid-base indicator dye, usually meta-cresol purple (mCP), to the seawater sample. The method was initially defined in the 1980s (Robert-Baldo et al., 1985; Byrne and Breland 1989), and the use of meta-cresol purple (mCP) as the indicator dye began The method was defined in the 1990s (Clayton & Byrne, 1993). The technique has been updated since then, and but it still lacks metrological traceability and reference materials (Ma et al., 2019; Carter et al., 2024a)."*

**Line 100:** Was this the same lot of dye used for all 11 cruises? Or were different batches of dye made over time, just with repeat purchases from the same brand?

*Only a few different batches of indicator dye from the same brand were used over the 11 cruises. For example, the same lot of indicator dye was used for the last four cruises. We have included it in Sect. 2.2.3 as follows: "Throughout the OVIDE-BOCATS program, the mCP dye used was from Sigma-Aldrich (Cat. No. 11,436-7 in the basic form; $C_{21}H_{17}NaO_5S$; molecular weight 404.41 g), with only a few different batches from this brand used over the 11 cruises."*

**Eq. 1:** Need to define what pK2 is for this equation. You should also make it clear that this pK2 is different from the pK2 defined in Clayton and Byrne (1993). They list their K1 and K2 as formation constants, whereas the K2 in this manuscript is a dissociation constant. Their K2 describes the formation of H2I, whereas the K2 in this manuscript describes the dissociation to form H+ and I2-. **Lines 171-172:** Again, make clear that the pK2 shown here is actually labeled as log K1 (so – pK1) in the CB'93 paper. I understand why you are showing yours as dissociation constants, but since you are referencing CB'93, you need to make it clear that they are not the same.

*Thank you for pointing this out. We agree, and have clarified it in the manuscript as follows: "The first three terms of Eq. (2) represent the second dissociation constant of mCP dye (pK$_2$, or -pK$_1$ in CB'93). The -pK$_1$ pK$_2$ reported by CB'93 is based on the TRIS (…)".*

**Lines 171-173**: This sentence is almost identical to a sentence in Lee et al. (2000) and should be properly cited, not just citing the last sentence of the paragraph.

*Done, thank you.*

**Line 178**: Would be useful to mention here that the mCP in CB'93 was also made in DI water. Which is different than what is done in this study (and described in detail later).

*We agree, and we have included the clarification as follows: "The CB'93 parameterization was developed using Kodak mCP dye, prepared in deionized water, which contained impurities contributing significant absorbance at 434 nm (referred to as $_{434}A_{imp}$)."*

**Line 185**: I think this value should be 0.018 from L'11.

*Corrected, thank you.*

**Line 205:** Since you're referencing eq. (11) of DB'17, you should note that your "Runpur" is noted as "Robs" in their paper.

*We agree, thank you. We have revised the text as follows: "Both the purified R values and their associated $_{434}A$ values were then used in Eq. (11) of DB'17 to compute the adjusted R values ($R_{unpur}$; referred to as $R_{obs}$ in DB'17) that reflect the contribution of mCP dye impurities ($_{434}A_{imp} \neq 0$) as follows: (…)".*

**Fig. 2:** It needs to be made clear that these 434Aimp values are specific to one lot from each of these vendors, which are listed in the CB'93 paper. These 434Aimp values aren't generic for all lots of impure mCP with the same vendor name.

*Yes, thank you. We have clarified this in the Figure 2 caption as follows: "(…) Each mCP dye is represented by a different color, with its corresponding $_{434}A_{imp}$ value (in units of $10^{-3}$ absorbance) specific to the lot used in DB'17, listed after the indicator dye name. Bold indicates mCP dye brands discussed in this work. All $_{434}A_{imp}$ values are taken from DB'17 and are specific to the lot used, except for Sigma-Aldrich, whose value was determined in this study."*

**Fig. 2:** What is the purified line showing? Is this using the same Runpur and comparing the CB'93 and L'11 parameterizations? It's difficult to follow the text in lines 220-222 describing this.

*Thank you for pointing out this confusion. We apologize, there was a typo in those lines. The 'purified' line shows the pH difference resulting from applying the two different parametrizations (CB'93 and L'11) to a purified mCP dye. No, it does not use the same Runpur; rather, the same Rpur is applied. In contrast, the other lines represent the difference between the pH computed using Runpur with CB'93 and Rpur with L'11. We have corrected the text as follows: "When the CB'93 parameterization is applied to $R_{unpur}$ ($_{434}A_{imp} = 0$) at S = 35 and 25ºC, the largest theoretical $pH_T$ differences (> 0.015 $pH_T$ units) are observed relative to $pH_T$ values obtained by applying the L'11 parametrization to the corresponding $R_{pur}$ under the same conditions (see turquoise line in Fig. 2)"*

**Line 219:** List the concentration for the mCP dyes determined in this study.

*We have included the final indicator dye concentration in the sample cell in the first and third experiment sections (not in the second one, since it is the same as in the first) to facilitate comparison with the concentration used by DB'17. A detailed response is provided to the comment below regarding "line 425".*

**Line 240:** Clarify that the blank measurement were recorded at the three target wavelengths (434, 578, 730 nm) like you do for the dye-addition measurements (line 246).

*We have clarified the text as follows: "For each sample, a blank measurement was performed after drying and cleaning both faces of the optical cell and placing it in the spectrophotometer's cell holder. Following blanking at the three target wavelengths (434 nm, 578 nm, and 730 nm) with sampled seawater, 75 µL of 2 mM mCP dye solution were added to each 28 mL sample cell using an adjustable repeater pipette (SOCOREX), resulting in a final mCP dye concentration of 5.36 µM in the cell. Dispenser syringes were wrapped in aluminum foil to prevent photodegradation of the mCP dye (Fontela et al., 2023). After the mCP dye addition, the cell was thoroughly shaken and placed back in the holder in the same orientation as for the blanking, and triplicate absorbance readings were carried out at the same target wavelengths as for the blank. All absorbance readings were carried out in the spectrophotometer's thermostatted cell compartment, maintained at 25.0 ± 0.2ºC."*

**Lines 249-252**: It should be clarified that measurements of R are insensitive to slight temperature changes (which is what Byrne and Breland 1989 noted). Also, Byrne and Breland were using cresol red, not mCP. However, pH is quite sensitive to temperature so accurate T measurements are necessary.

*Thank you for the clarification. We have rewritten the text as follows: "Byrne & Breland (1989) demonstrated that  R  measurements are largely insensitive to small temperature variations, for cresol red dye. The same general behaviour applies to mCP dye, in which R values between 0.5 and 2. exhibit pH_T errors of less than 0.001 pH_T units per 0.5°C change when using the  L'11 parameterization. This insensitivity arises because, for mCP, the temperature dependence of the indicator dye's pK_2  closely parallels  the temperature dependence of seawater pH. The CB'93 pH_T parametrization shows a slightly greater temperature sensitivity, such that temperature deviations must be kept within approximately ±0.5°C to limit the pH_T error to ≤0.0012 pH_T units. Therefore, , it is recommended to ensure that temperature deviations remain within ±0.5°C of the reference temperature (25°C). In our procedure, sample temperature was monitored every five measurements to verify that it remained within this tolerance "*

**Line 264:** Specify that an R of 1 corresponds to a pHT of 7.67 at what salinity and temperature?

*We have specified this as follows: "(…) R values remained close to 1, corresponding to a pH_T of approximately 7.67 (Li et al., 2020; at S = 35 and T = 25ºC)."*

**Line 279:** Can authors justify why their double-dye additions used 50 uL increments instead of 75 uL like their experiments? The authors account for the difference correctly in their equation 6, but I think readers will wonder the reasoning.

*The 75 µL addition corresponds to the center or intermediate range of the deltaR adjustment. Using 50 µL increments allowed us to examine the indicator dye effect both "behind" and "in front" of the 75 µL reference point, providing a clearer understanding of the response while still correctly accounting for the difference in Eq. 6. We have justified this in the manuscript as follows: "Following blanking, an initial addition of 50 µL of mCP dye solution was made to each sample, and absorbance was measured as described in Section 2.2.2. A second addition of 50 µL of mCP dye solution was then made (resulting in a total of 100 µL of mCP dye solution added), and absorbance measurements were repeated. These double-addition experiments enabled the determination of linear regressions of the change in $pH_T$ ($\Delta pH_T = pH_{T,2}$ - $pH_{T,1}$; where subscripts 2 and 1 refer to 100 µL and 50 µL of mCP dye solution added, respectively) or R ($\Delta R = R_2 - R_1$) as a function of the initial $pH_{T,1}$ or $R_1$, respectively (Supporting Information Fig. 1). This two-step 50 µL addition bracketed the typical 75 µL reference volume added to the sample, allowing us to evaluate the dye effect on the $\Delta R$ or $\Delta pH_T$ both below and above this reference. The corresponding relationship was expressed as: (...)".*

**Line 297**: Might be useful to add a clarifying sentence here to explain that at pHT y=0, the R (pH) of the original sample and the R (pH) of the indicator are the same, which is why no change is observed.

*Thank you. We have clarified the text as follows: "At $pH_T^{y=0}$, the R ($pH_T$) of the original sample and the R ($pH_T$) of the indicator dye are the same, so no change is observed. If $pH_m > pH_T^{y=0}$, then (...)".*

**Line 307**: For a couple of the cruises, normalizing the delta R with the isosbestic point reduced the R2. Can the authors explain why that would be?

*We apologize for the confusion. Upon checking the computations, we identified a few typos when transferring the data for the 2018 cruise, which have now been corrected. Fortunately, these do not affect the equations applied to the data (last column). In contrast, the 2014 cruise equations are correct, and the slightly lower $R^2$ for some cases arises from an anomalous average of the indicator dye effect perturbation at high pH values.*

**Line 339**: R=1 (or pHT=7.65) at what temperature and salinity?

*We have clarified this as follows: "In contrast, deviations from R = 1 (or $pH_T$ = 7.65; at S = 35 and T = 25ºC) enhance this (...)".*

**Fig. 3:** Also need to specify which temperature condition.

*Done, thank you.*

**Lines 368-369:** How did the researchers determine when the volume deviated by more than 20%?

*We determined deviations by monitoring the absorbance at 488 nm (measured or computed). For each cruise, the mean A488 was computed across all samples measured in the field layer*

*(approximately 2,000 per cruise), providing a robust reference value. Any sample whose A488 deviated by more than 20% from this mean was considered to exceed the 20% threshold. This procedure was repeated for all 11 OVIDE-BOCATS cruises. We have clarified in the text as follows: "During the OVIDE-BOCATS cruises, mCP dye was manually added to samples using an adjustable repeater pipette (see Section 2.2.2). For each cruise, volume deviations associated with manual addition were assessed by comparing each sample to the cruise-specific mean $_{488}A$, computed from all field-layer samples (~2000 per cruise). Manual addition*  *resulted in volume deviations exceeding 20% in approximately 3% of the samples (~ 706 cases), potentially affecting $\Delta R$ and $\Delta pH_T$ determinations. To address variability in the volume of mCP dye solution added, (...) ".*

**Lines 395-396:** Authors should note that some uncertainty is introduced by collecting duplicates on 2 different Niskin bottles, rather than 2 samples from the same Niskin. (i.e., small leaking in one bottle, biological activity, delay in closing between Niskins, etc.)

*We have clarified this as follows: "Throughout the 11 OVIDE-BOCATS cruises, a total of 502 duplicate samples were collected to evaluate the reproducibility of $pH_T$ measurements using an unpurified*  *mCP dye. At selected stations, two Niskin bottles were closed at the same depth to obtain replicates. Any uncertainty introduced by collecting duplicates on two different Niskin bottles (e.g., small leaks, biological activity, or delay in closing) was neglected. Figure 4 displays the absolute $pH_T$ differences between replicates for each cruise. The overall mean and standard deviation of these differences is 0.0014 ± 0.0015 $pH_T$ units (N = 502)."*

**Fig. 4:** The grey shading is useful but hard to see. Can the authors make it slightly darker? Min and max values are also very small and hard to read.

*Done, thank you; and we applied the same recommendations to Fig. 8.*

**Line 425:** Would be useful to include the final dye concentration in the sample cell for the 2 dye types for a direct comparison, since the experiments used different volumes and stock indicator concentrations.

*Thank you for the suggestion. We agree and have included the final indicator dye concentration in the sample cell in the first and third experiment sections (not in the second one, as it is the same as in the first) as follows:*

*"3.1 TRIS buffer validation*

*(...) Multiple bottles from both batches were measured during the BOCATS2-2021 and BOCATS2-2023 cruises using two mCP dye solutions: (i) unpurified mCP (75 µL of 2 mM solution; Sigma-Aldrich; 5.36 µM final mCP dye concentration in the sample cell), and (ii) purified mCP (10 µL of 11 mM solution; provided by Prof. R. Byrne's laboratory; 3.93 µM final mCP dye concentration in the sample cell). (...)*

*3.3 Duplicate measurements of modified seawater*

*(...) For each batch, four to six samples were analyzed for pH$_T$ using the standard OVIDE-BOCATS procedure (CB'93+DVD'98 and 75 μL of unpurified Sigma-Aldrich mCP dye [2 mM] solution added to the cell; 5.36 μM final mCP dye concentration in the sample cell), and an equivalent number was measured using purified mCP dye (75 μL of mCP dye [2.5 mM] solution added to the cell; 6.70 μM final mCP dye concentration in the sample cell; FB5-2017 from Dr. Byrne's laboratory and the L'11 parametrization). (...) ".*

**Fig. 5:** Might be useful to reorganize the figure so that the purified dye column is on the right-most side. The unpurified dye columns could then all be shown together as increasingly improving the results to most closely match the purified dye results.

*Done, thank you.*

**Fig. 6:** The "N=176" within the figure seems unnecessary, since it's listed in the figure caption. Its placement is confusing since it makes it appear that the "N=176" is specifically referring to the middle column of the figure, rather than all of the data.

*Removed from the figure, thank you.*

**Lines 514-516:** Mention this finding is reasonable since you are working with two different lots of dye. The phrasing now makes it seem like you are recommending a change to the DB'17 value, rather than that you have a different 434Aimp value because you have a different lot of dye.

*Thank you, we agree. We have rephrased the phrase as follows: "DB'17 proposed a value of $_{434}A_{imp}$ = 0.004413 absorbance units for a Sigma-Aldrich mCP dye at a final concentration of 3.3 μM in the sample cell ($_{488}A$ = 0.225; Takeshita et al., 2021). Based on our experimental results, we found that a better fit under our conditions corresponded to a $_{434}A_{imp}$ value of 0.0034 absorbance units, which is reasonable given that we are working with a different lot of mCP dye."*

**Line 543:** No pH values were flagged as 4? If you're going to highlight the number of flag 3, you should include the number of other flags as well.

*Thank you for noticing this. We have identified a mistake in the generation of the files in different formats for upload to the repository, in which some pH values were inadvertently removed. This occurred when one or some marine carbonate system parameters (pH, total alkalinity, dissolved inorganic nutrients, etc.) had a flag 4, which prevented our conversion routines from resolving the seawater CO$_2$ system and transforming pH values to in-situ conditions. Only complete pairs with pHT at 25ºC and in-situ were previously included, resulting in the removal of some data. We have fixed this issue to allow the conversion for all values, and re-uploaded the files in the repository. Moreover, from the new total of 23,843 samples, 23,773 were flagged as good (2), 19 as questionable (3) and 51 as bad (4). We have corrected this number in the manuscript, and this text as follows: " The dataset incorporates a quality flagging scheme consistent with GLODAPv2 recommendations (Key et al., 2015; Olsen et al., 2016), where flag 2 denotes good data (23,773*

*samples)*, *3 (19 samples) and 4 (51 samples) indicate questionable and bad data, respectively, and 9 denotes not measured.* "

*In addition, we have updated the DOI to the new corrected version in Sect. 6 as follows:* " *The complete OVIDE-BOCATS pH$_T$ dataset presented in this study is made available at*  *https://doi.org/10.5281/zenodo.17141183 (Pérez et al., 2025) in multiple formats to ensure broad accessibility and compatibility with different research workflows.*"

**Lines 545-547**: So, were the pHT data from 2002-2018 not corrected using the DB'17 adjustment and L'11 equation? Or this was what was in GLODAPv2 before and will be updated using DB'17 and L'11 for v3? It needs to be very clear which data is in each version. Same for the 2021 and 2023 cruises – which procedure was used and what is the final product? Spell it out here for readers who skim, since this is a very important point.

*We fully agree on the importance of clarifying this point and thank Reviewer #2 for highlighting it. We have revised the text as follows:* "*Spectrophotometric pH$_T$ data collected between 2002 and 2018, available in the GLODAPv2.2023 release (Lauvset et al., 2024), were computed using the CB'93 parameterization with the DVD'98 correction (+0.0047 pH$_T$ units)**. The newly compiled pH$_T$ dataset presented here updates and corrects these data following the procedure presented in Sect. 3.4, applying the DB'17 adjustment and L'11 equation. In addition, the dataset significantly extends the temporal coverage by including pH$_T$ measurements from the 2021 and 2023 cruises, which were not previously available; these measurements are also corrected using the same procedure, resulting in a consistent final product across all cruises.*"

**Line 603**: What's the uncertainty in these saturation depth changes? I would be curious what is the range in the depth that saturation = 1 based on the uncertainty in pH measurements themselves?

*The uncertainty in the saturation depth change due to the uncertainty in pH measurements (given the reproducibility of the doubloons and the STD of the average pH$_T$ in the NEADW layer over 11 cruises) is 17 meters, and for Ω$_{Ar}$ 0.0033 units. We have included in the manuscript as follows:* "*This reevaluation reveals a more pronounced reduction in aragonite saturation at the surface (from -0.040 to -0.065), relative to pre-industrial conditions, which progressively diminishes with depth, reaching changes of -0.016 near the seafloor. Although changes at depth appear small in absolute terms, the weak vertical gradient in aragonite saturation in deeper layers translates into a significant vertical shift in the saturation horizon—rising by approximately 120 m to 200 m. Based on the reproducibility of the doubloons (Sect. 2.2.5) and the standard deviation of the mean pH$_T$ in the NEADW layer over 11 cruises (Sect. 4.2), the uncertainty in the saturation-depth change is estimated at 17 meters, while the uncertainty in the aragonite saturation state is 0.0033 units. For instance, in the NA subpolar gyre, where the aragonite saturation horizon currently resides (...)* "

**Fig. 9:** Difficult to read pH scale numbers and numbers in each panel, also difficult to read the x and y axes labels and numbers.

*Corrected, thank you.*

**Lines 649-651:** What are the uncertainties on these pH rate changes per year?

*We have addressed this by adding a new figure to the Supporting Information (Supporting Information Fig. 7) that shows the standard deviation of pH$_T$ across the 11 cruises (panel A) and the standard error of the pH$_T$ rate (panel B). The text has been updated to guide the reader to this figure as follows: "These upper layers also exhibit high interannual pH$_T$ variability ( Supporting Information Fig. 7), which correlates negatively with AOU (Supporting Information Fig. 6b)."*

Technical Corrections

**Line 14:** Provide the acronym meanings for GO-SHIP and OVIDE-BOCATS. *Done*

**Line 97:** Provide acronym meaning for CLIVAR. *Done*

**Line 163**: Should be CB'93. *Done*

---

## Author Comment (AC3)

**Anonymous Referee #3,** 08 Jan 2026

General comment:

In my opinion the submitted manuscript "Two decades of pHT measurements along the GO-SHIP A25 section" is excellent and I suggest should be accepted after minor revisions.

Measurements of any parameter to high accuracy and precision over a long timescale are challenging especially where no certified reference material is available. Further confounding the challenge is the evolving consensus in best practice. The attractive simplicity of spectrophotometric seawater pH measurements has been long established, however the various improvements and refinements potentially thwart production of a consistent high quality long-term data set.

This paper summarises the fundamentals and the issues with dye impurity well then proceeds to discuss various replication and quality assurance protocols used to eventually ascribe a justifiable offset with uncertainty between pHT data calculated from absorbances determined with unpurified and purified dye stocks. It is difficult to imagine anything further that could have been done ensure the data quality.

*We would like to thank Anonymous Referee #3 for the thoughtful review, the positive assessment of our manuscript, and for recognizing the challenges of producing high-quality long-term spectrophotometric seawater pH measurements, particularly in the absence of certified reference materials and given the evolution of best practices over time. All comments have been carefully considered, as detailed in the responses below.* Anonymous Referee #3's evaluation is reproduced in black, *the author's responses appear in green and italics, the original manuscript text appears in black and italics, and* the changes introduced in the manuscript are shown in blue and italics.

Comments:

**C1:** There are a lot of acronyms used and not all are explained on their first use**.**

*Thank you for pointing this out. We have provided the acronym meanings for GO-SHIP (line 14), OVIDE-BOCATS (line 14) and CLIVAR (in line 97).*

**C2: Line 163** CB'13 should be CB'93

*Corrected, thank you.*

**C3: Line 192** expansion of DB'17.

*This is explained previously in line 93 as follows: " (…) and correct them accordingly (Douglas & Byrne, 2017; hereafterDB'17; (…) ". However, if the Referee considers it appropriate, we could expand on this in the line 192.*

**C4: Line 193** language could be improved

*Thank you, agree. We have corrected the line as follows: "This approach allows  R to be corrected for the contribution of impurities  at $_{434}A$ (i.e., $_{434}A_{imp}$), and consequently (…) ".*

**C5: Line 337** orange squares in Fig. 3

*Thank you, corrected.*

**C6: Line 410** Font sizes, point colour makes chart difficult to when printed.

*Thank you for noting this. We have increased the font sizes and adjusted the point colors to prevent them from appearing faint when printed.*

**C7: Line 447** Colours faint when printed. Could authors comment on why the σ on the purified dye differences (0.0014) seems notably smaller than the unpurified dye (0.0024) for Tris but not modified seawater (both 0.0017).

*Thank you for pointing this out. We have updated the color map to improve contrast and visibility when printed, including black-and-white reproduction.*

*Regarding the difference in σ, given the mean reproducibility of our measurements (~0.0015 $pH_T$ units), we agree that the σ obtained with the unpurified dye is slightly higher than expected. We explored the data, and a couple of measurements with the unpurified dye exhibit a positive bias together with a slight temperature bias.*

**C8:** Line 591 upper-right insert.

*Corrected, thank you.*

**C9:** Couldn't see the dataset due to restricted access in Zenodo

*We sincerely regret that you were unable to access the dataset. At the time of manuscript submission, and following the options provided by ESSD, we chose to keep the dataset in a permanent repository (Zenodo, with DOI) under embargo until publication. In parallel, we created a temporary public repository containing the identical dataset for reviewers to use during the review process. This temporary link is: https://saco.csic.es/s/kKqDXFYGKsKbaXj. According to ESSD procedures, this link should have been sent upon acceptance of the referees. We apologize for the oversight and any inconvenience it may have caused.*

---

## Author Response (AR2)

Dear Topic Editor,

Thank you very much for handling our manuscript. We are grateful for the acceptance of our manuscript and are pleased to submit the revised version incorporating your minor corrections.

Please find attached the revised manuscript, which now includes the DOI of the final dataset in the abstract. We have double-checked it, as well as in Section '6. Data availability', and the corresponding reference. In addition, in accordance with the editorial ESSD guidelines, we have revised and corrected the references list and corrected the numbering and citations of the Supplementary Material.

We would like to thank you once again for your time and the constructive comments provided.

Kind regards,
 on behalf of all coauthors,

Marta López Mozos